# Ontology-based taxonomical analysis of experimentally verified natural and laboratory human coronavirus hosts and its implication for COVID-19 virus origination and transmission

Yang Wang[1,2,3], Muhui Ye[4], Fengwei Zhang[1], Zachary Thomas Freeman[3], Hong Yu[1,2], Xianwei Ye[1,2]*, Yongqun He[3,5,6]*

1 Guizhou University School of Medicine, Guiyang, Guizhou, China, 2 Department of Respiratory and Critical Care Medicine, Guizhou Provincial People's Hospital and NHC Key Laboratory of Immunological Diseases, People's Hospital of Guizhou University, Guiyang, Guizhou, China, 3 Unit for Laboratory Animal Medicine, University of Michigan Medical School, Ann Arbor, MI, United States of America, 4 Chinese University of Hong Kong (Shenzhen), Shenzhen, Guangdong, China, 5 Department of Microbiology and Immunology, University of Michigan Medical School, Ann Arbor, MI, United States of America, 6 Center for Computational Medicine and Bioinformatics, University of Michigan Medical School, Ann Arbor, MI, United States of America

* yxw1205@163.com (XY); yongqunh@med.umich.edu (YH)

**Data Availability Statement:** All relevant data used in our study have been added to the paper and its Supporting Information files.

## Abstract

To fully understand COVID-19, it is critical to study all possible hosts of SARS-CoV-2 (the pathogen of COVID-19). In this work, we collected, annotated, and performed ontology-based taxonomical analysis of all the reported and verified hosts for all human coronaviruses including SARS-CoV, MERS-CoV, SARS-CoV-2, HCoV-229E, HCoV-NL63, HCoV-OC43, and HCoV-HKU1. A total of 37 natural hosts and 19 laboratory animal hosts of human coronaviruses were identified based on experimental evidence. Our analysis found that all the verified susceptible natural and laboratory animals belong to therian mammals. Specifically, these 37 natural therian hosts include one wildlife marsupial mammal (i.e., Virginia opossum) and 36 Eutheria mammals (a.k.a. placental mammals). The 19 laboratory animal hosts are also classified as therian mammals. The mouse models with genetically modified human ACE2 or DPP4 were more susceptible to virulent human coronaviruses with clear symptoms, suggesting the critical role of ACE2 and DPP4 to coronavirus virulence. Coronaviruses became more virulent and adaptive in the mouse hosts after a series of viral passages in the mice, providing clue to the possible coronavirus origination. The Huanan Seafood Wholesale Market animals identified early in the COVID-19 outbreak were also systematically analyzed as possible COVID-19 hosts. To support knowledge standardization and query, the annotated host knowledge was modeled and represented in the Coronavirus Infectious Disease Ontology (CIDO). Based on our and others' findings, we further propose a MOVIE model (i.e., Multiple-Organism viral Variations and Immune Evasion) to address how viral variations in therian animal hosts and the host immune evasion might have led to dynamic COVID-19 pandemic outcomes.

**Funding:** This research was supported by Youth Found of Guizhou Provincial People's Hospital of China, GZSYQN[2019]09, Guiyang Science and Technology Bureau Science and Technology Major Special Plan, Guizhou Provincial People's Hospital Public Health and Epidemic Prevention and Control Series Research Contract [2020] -4-1, the Non-profit Central Research Institute Fund of Chinese Academy of Medical Sciences 2019PT320003, and a bridge fund (to YH) at the Unit for Laboratory Animal Medicine in University of Michigan Medical School.

**Competing interests:** The authors have declared that no competing interests exist.

## Introduction

Zoonotic coronaviruses have caused dramatic impacts on humans. The existing COVID-19 pandemic has caused a disaster in public health worldwide. SARS-CoV-2, the cause of COVID-19, is a coronavirus that infects humans and leads to severe acute respiratory syndrome in humans. By 2 November 2023, COVID-19 had caused 771,679,618 confirmed cases of COVID-19, including 6,977,023 deaths, as reported to WHO (https://covid19.who.int/). In addition to SARS-CoV-2, two other coronaviruses also caused major losses in this century. In 2002, Severe Acute Respiratory Syndrome (SARS) emerged in China and eventually caused 8,098 confirmed human cases in 8 months and 774 deaths in 29 countries [1,2]. In 2012, the Middle East Respiratory Syndrome Coronavirus (MERS-CoV) outbreaks, initially found in Saudi Arabia [3], resulted in 2,260 cases and 803 deaths across 27 countries [4,5]. In addition, four other human coronavirus strains, including HCoV-229E, HCoV-NL63, HCoV-OC43, and HCoV-HKU1, were also found worldwide and caused common cold in humans [6]. Understanding how these human coronaviruses spread from host species to humans is crucial to preventing a pandemic.

The human coronaviruses are likely able to spread and transmit from animals to humans. These coronaviruses appear to break down the species barrier through the transmission of the natural host and likely intermediate host and achieve the replication of the virus in the human body. Previous studies have found many natural and laboratory animal hosts of human coronaviruses, such as bat [7], civet [8], camel [9], deer [10], monkey [11], and mouse [12]. However, the exact scope of the human coronavirus hosts and their transmission relations remain unclear.

In the informatics domain, ontology is a structured vocabulary that represents entities and relations among the entities in a specific domain using a human- and computer-interpretable format [13]. Many biological and biomedical ontologies have been developed and widely used. For example, the NCBITaxon ontology [14] is a taxonomy ontology developed based on the classification of various types of cellular organisms and noncellular self-replicating organic structures including viruses in the NCBI taxonomy database. The NCBI Taxonomy system provides a way to search the taxonomical information of specific animal types. However, it does not provide an automatic way to extract the hierarchical and integrative taxonomical information from a group of animal types. To address this issue, we could utilize the NCBI-Taxon ontology, which is derived from the NCBI Taxonomy system, and a specific tool (such as Ontofox [15] as used in this paper) to extract a small subset of the taxonomical hierarchy of related animals in an automatic and efficient way. The Coronavirus Infectious Disease Ontology (CIDO) is a community-based ontology that systematically represents various coronavirus-related topics, including etiologies, hosts, transmissions, diagnosis, drugs, and prevention [16–20]. By systematically incorporating COVID-19 knowledge in CIDO, we are able to develop more advanced applications, such as data standardization and integration, better mechanistic understanding of virulence and transmission, natural language processing (NLP) for clinical and basic mechanism research, and machine learning and drug cocktail design [16–18].

This study aims to survey and identify all verified hosts of human coronaviruses based on literature and data collection, followed by systematic analysis of these human coronavirus hosts using ontology-based taxonomical classification and bioinformatics methods, with the ultimate goal of studying COVID-19 viral origination and transmission mechanism. Given the importance of COVID-19, we have focused our study on the hosts of SARS-CoV-2. Our study found that all verified natural and laboratory animal hosts of human coronaviruses belong to therian mammals. Genetically modified mouse models appeared to be more susceptible to

virulent human coronavirus infections developing similar clinical symptoms, suggesting that host susceptibility depends on many factors including genetic modification for coronavirus binding to host cells. The increased viral virulence after serial passages in mice suggested a path of viral origination and eventual transmission to human. The original Huanan Seafood Wholesale Market animals identified early in the COVID-19 outbreak were analyzed with the goal to identify possible intermediate hosts. Bioinformatical approaches, including calculated changes in energy (ΔΔG) of the SARS-CoV-2 S-protein:ACE2 complex binding and phylogenetic analysis, were also applied to explore possible host origin and transmission mechanism. The learned knowledge is further modeled and represented in the CIDO ontology, supporting integrative knowledge representation and reasoning. Furthermore, based on our and others' findings, we propose an integrative MOVIE model, standing for "Multiple-Organism viral Variations and Immune Evasion", to address how the COVID-19 originated and transmitted among therian mammal hosts.

## Methods

### Collection and annotation of verified hosts of human coronaviruses

Instead of generating new data, this study began by identifying verified animal hosts of human coronaviruses from existing literature. Peer-reviewed journal articles from PubMed were mined and annotated to identify various hosts for different human coronaviruses, including natural hosts and laboratory animal models with experimental evidence. The PubMed searching keywords commonly used include: (SARS-CoV OR SARS-CoV-2 OR MERS-CoV OR "human coronavirus") AND host. Meanwhile, other resources including WHO reports and trusted newspapers are also searched. In addition to SARS-CoV, SARS-CoV-2, and MERS-CoV, four human coronaviruses cause the common cold, which includes HCoV-229E, HCoV-NL63, HCoV-HKU1, and HCoV-OC43. The reported evidence for being a host for a human coronavirus was then extracted, annotated, and recorded in a pre-designed Excel file. The recorded information was also summarized in formal tables and provided in the manuscript.

To be determined as a host of human coronaviruses, the required evidence is supposed to include at least one experimental confirmation using methods such as virus isolation, genomic sequencing, RT-PCR, and antibody neutralization assay. Clinical evidence such as related symptoms recorded but not required for inclusion in our collection. This evidence is applied to wild-type animals in its natural situation only, and transgenic animals are not counted. Although the transgenic mouse model is a much more effective model for human coronavirus study, wild-type mice would also be infected with some human coronaviruses such as SARS-CoV-2 (B.1.351) [21] and MERS-CoV [22]. Therefore, the mouse is considered a natural host as well as a laboratory host.

If the required evidence as described above is not met, a suspected animal is not considered a host. For example, Wikipedia contains a web page that lists animals that can get SARS-CoV-2 [23]. After careful examination, three animals listed on Wikipedia, including swan, zebrafish, and housefly, were not included in our verified list of SARS-CoV-2 hosts because the evidence provided for these animals could not be traced and verified. For instance, the swan was cited in the WHO report [24]. However, a careful examination of the report and its cited resource could not find the required evidence of including swans as a host of SARS-CoV-2. Housefly, an insect, was also found to contain the SARS-CoV-2 virus [25]. However, the houseflies likely took up the blood fluids of the COVID-19 patients and virus RNA survived in the insect body without replicating, which means that the houseflies serve as a vector instead of the host. This hypothesis was further confirmed by another independent study [26]. Zebrafish is a kind of

vertebrate that share a high degree of sequence and functional homology with mammals. However, zebrafish is not included as a laboratory animal model because zebrafish has not been found effectively infectable with human coronaviruses, probably due to dissimilarities between human and zebrafish ACE2 in the Spike-interaction region [27]. Natural exposure or microinjection in different anatomic locations, including the coelom, pericardium, brain ventricle, or bloodstream, led to a quick decrease of SARS-CoV-2 RNA in wild-type zebrafish larvae. After inoculation in the swim bladder (an aerial organ sharing similarities with the mammalian lung), the detected coronavirus decreased within 24 hours and then became stable through qRT-PCR, however, no clear evidence for the production of new SARS-CoV-2 virions was observed [27], suggesting the failure of achieving detectable infection even in the swim bladder. A mosaic overexpression of hACE2 was not sufficient to achieve detectable infectivity of SARS-CoV-2 in zebrafish embryos or in zebrafish cells in vitro [27], further justifying the exclusion of zebrafish as a laboratory human coronavirus host. However, further studies revealed that the humanized zebrafish, xeno-transplanted with human lung epithelial cells, could be a good model for SARS-CoV-2 infection [28]. Therefore, houseflies, swans, and zebrafish are not included in our host list.

An additional effort was also used to identify genetically modified mouse models used for the study of human coronaviruses. A meta-analysis of those mouse models susceptible to the infection of SARS-CoV, MERS-CoV, and SARS-CoV-2 was also conducted with careful information annotation and analysis.

## Ontology-based taxonomical classification of human coronaviruses and their hosts

The host taxonomy analysis was performed in an ontology-based approach, i.e., by using the NCBI Taxonomy Ontology (NCBITaxon). The NCBI taxonomy database provides the taxonomical structure of approximately 1 million taxonomical terms, which is difficult to extract a subset of animal species and form a hierarchical structure of these species. After we transfer the NCBI taxonomical IDs of human coronaviruses and their hosts to NCBITaxon ontology IDs, we used the tool Ontofox [15] to efficiently generate a subset of the taxonomical species, their related ancestors, and the semantic relations among these taxonomical terms. The hierarchical structure of these terms was then displayed using the Protégé-OWL editor [29].

For Ontofox to run, the NCBI Taxonomy ontology IDs of specific species were used as the input. The Ontofox choice of "includeComputedIntermediates" was used to compute and retrieve the closest ancestors of different species in the hierarchical taxonomical tree. In addition to the different levels of ancestors, the Ontofox tool can also extract the relations among different levels of taxonomical terms and specific annotations such as scientific species names (i.e., labels) and common names (e.g., synonyms). The Ontofox output files were opened using the Protégé-OWL editor for visualization of the hierarchical structure and annotations. Screenshots of such visualization of human coronaviruses, nature hosts, and laboratory models, which display the relationships among different taxonomical terms, were finally generated and saved as images.

## Ontological and computational analysis of Huanan Seafood Wholesale Market animals

To further trace the origin animal species of COVID-19 infections, we analyzed all animals from the Huanan Seafood Wholesale Market, which is frequently considered the first place of the COVID-19 outbreaks in Wuhan, China. The World Health Organization (WHO) convened a joint WHO-China study during 14 January– 10 February 2021 and provided a report

titled *WHO-convened global study of origins of SARS-CoV-2*: *China Part* [30]. This report lists all the animal species identified in the Wholesale Market and provides the degree of the susceptibility of individual animals to COVID-19 infection. In our study, we mapped these animals to NCBITaxon ontology term IDs, which were then used as input for our Ontofox tool analysis to extract the taxonomical hierarchical relations of these animals. Since SARS-CoV-2 S-protein: ACE2 complex binding is a good indicator of being SARS-CoV-2 host, we compared the WHO-listed susceptibility results with the calculated changes in energy (ΔΔG) of the SARS-CoV-2 S-protein:ACE2 complex binding as previously studied by Lam, et al [31]. Lower ΔΔG values represent more stable binding, and therefore higher risk of infection. The ΔΔG cutoff of 3.7 was used. Those species with a ΔΔG lower than 3.7 were considered as at risk of infection, and otherwise considered as a low risk of infection [32].

### ACE2 phylogenetic analysis method

A phylogenetic analysis was performed to establish the phylogenetic relations among different hosts of human coronaviruses using a method previously reported [33]. Specifically, the ACE2 protein sequences from different host species of human coronaviruses were found from the NCBI Protein Database (https://www.ncbi.nlm.nih.gov), and then aligned using the Muscle program within the MEGA software [34]. Those animals without ACE2 record included on the website were not included in this phylogenetic analysis. The phylogenetic tree of these proteins was generated with the META tool using a Maximum likelihood method [35]. The phylogenetic tree was displayed using the Interactive Tree Of Life (https://itol.embl.de), an online tool used for displaying, manipulating, and annotating phylogenetic and other trees [36].

### CIDO ontological representation and analysis of human coronavirus-host relations

To support data standardization, integration, and analysis, we used the Coronavirus Infectious Disease Ontology (CIDO) [16,18,37,38] to ontologically model, represent, and analyze the relations between human coronaviruses and hosts. The eXtensive Ontology Development (XOD) strategy [39] was applied for the CIDO ontology development. Specifically, new CIDO design patterns and axioms were first generated to semantically link different terms including the coronaviruses and their hosts. The Ontorat tool [40] was then used to transform the input data of human coronaviruses and their related hosts collected in Excel to OWL format for further display, editing, and analysis with Protégé-OWL 5.5 editor [41]. To demonstrate the usage of the CIDO representation, the SPARQL RDF query language was used to query the Ontobee triple store that contains the CIDO knowledge represented as "subject-predicate-object" triples [42].

## Results

### Taxonomical classification of various human coronaviruses

Fig 1 shows the taxonomical classification of seven human coronaviruses, together with the Infectious bronchitis virus (IBV) (an avian coronavirus as control), and their relations under the hierarchy of taxonomy. Specifically, coronaviruses are positive-stranded RNA viruses, belonging to the order Nidovirales, family Coronaviridae, and subfamily Coronavirinae. The subfamily Coronavirinae contains the four genera Alpha-, Beta-, Gamma-, and Deltacoronavirus. SARS-CoV and SARS-CoV-2 are members of the Sarbecovirus subgenus under the genus Betacoronavirus. MERS-CoV falls in Merbecovirus under the same genus Betacoronavirus (Fig 1) [43]. The four coronaviruses that cause the common cold are also under the

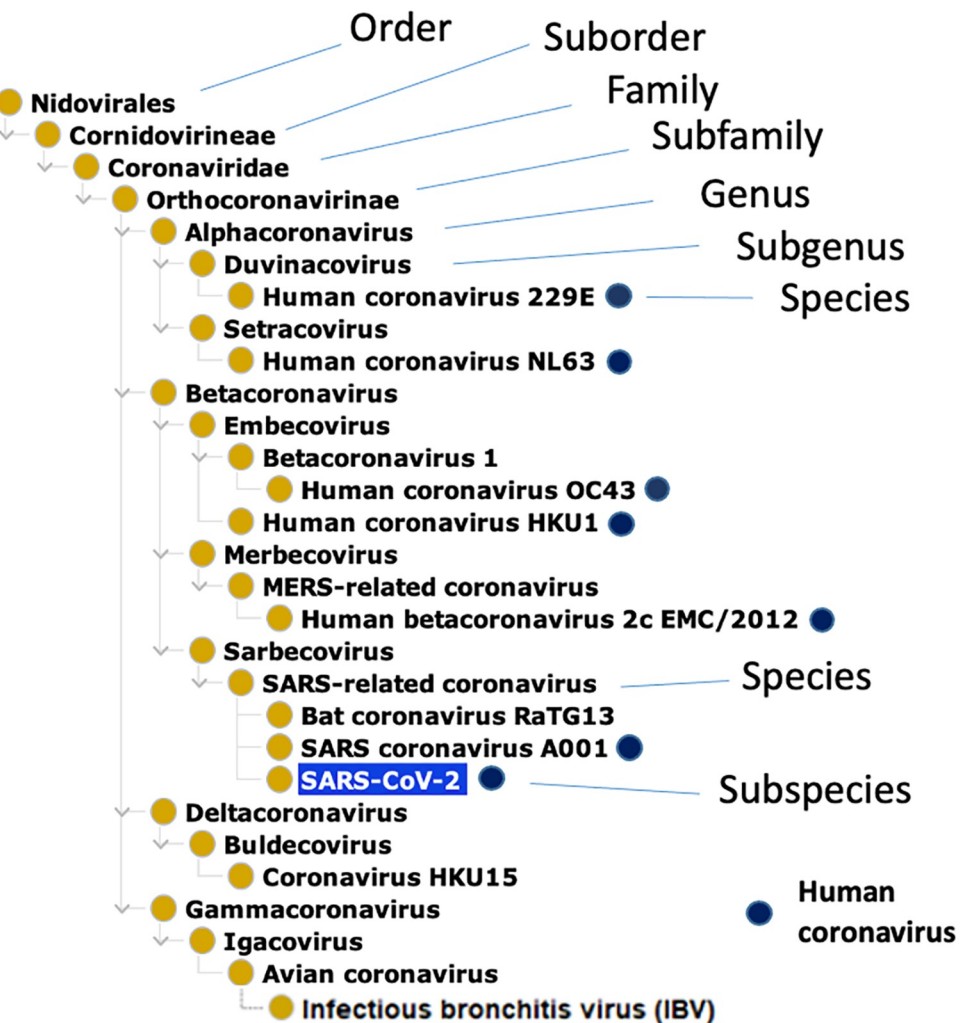

**Fig 1. Taxonomical hierarchy of human coronaviruses based on the NCBITaxon ontology.** The dark blue circles label those coronaviruses capable of infecting humans. Under the subgenus Sarbecovirus and genus Betacoronavirus, SARS–related coronavirus is a species that includes specific SARS-CoV-1 and SARS-CoV-2 strains. Given that there are many specific strains of MERS-CoV and SARS-CoV species, only representative strains are shown here. Bat coronavirus RaTG13 (a bat coronavirus strain highly homologous to SARS-CoV-2), Coronavirus HKU15 (a coronavirus species that infects pigs), and Avian coronavirus (a coronavirus species that infects birds) are also included here as examples of non-human coronaviruses.

subfamily of Orthocoronavirinae. Specifically, HCoV-229E is a member of the Duvinacovirus subgenus, HCoV-NL63 is a member of the Setracovirus subgenus, and they are both under the Alphacoronavirus genus. HCoV-HKU1 and HCoV-OC43 are members of the Embecovirus subgenus under the genus Betacoronavirus. The IBV avian coronavirus, which does not infect humans, is a member of the Gammacoronovirus (Fig 1).

While this study aims to systematically analyze verified natural and laboratory human coronavirus hosts, the taxonomical analysis of these human coronaviruses allows us to know how these human coronaviruses are closely related, which provides a basis for our further analysis of the hosts of these human coronaviruses. Indeed, the taxonomical classification of the coronaviruses appears to be associated with the host species that these coronaviruses turn to infect. In general, Alpha- and Betacoronaviruses mainly infect mammalian species including humans,

and Gamma- and Deltacoronaviruses primarily infect birds [6]. Note that although bats can fly, it is a mammalian species. Bat-borne betacoronaviruses are closely related and responsible for many human respiratory infections [44].

## Identification and classification of 37 verified natural animal hosts of human coronaviruses

In this article, natural hosts of human coronaviruses are defined as those animals identified to be infected with human coronaviruses with experimental evidence. Table 1 collects 37 natural hosts of different types of human coronaviruses based on convincing experimental evidence as reported in the literature. All these animals are under Theria <mammals>, i.e., therian mammals (Fig 2).

These 37 therian mammals include a wildlife marsupial mammal (i.e., *Didelphis virginiana*) and 36 Eutheria mammals (a.k.a. placental mammals) (Fig 2). *Didelphis virginiana* is also called Virginia opossum, which belongs to the Didelphimorphia order under the Metatheria clade. Metatherian mammals, also known as marsupials, are an ancient group, and most marsupials are found in Australasia (around 200 species) and Central and South America (around 70 species). A study conducted by Goldberg et al. found 8% of wildlife Virginia opossums having detectable serum antibodies against SARS-CoV-2 [51].

Eutheria is synonymous with placental mammals [52]. In the taxonomical classification, most of our verified natural hosts are categorized within the superorders Euarchontoglires and Laurasiatheria under the clade Boreoeutheria, and only two (i.e., large hairy armadillo and giant anteater) under the superorder Xenarthra. Euarchontologilres include Homininae and Murinae (rodents) subfamilies (Fig 2). The Homininae subfamily contains *Homo sapiens* (humans) and gorillas. In addition to humans, the world's first positive case of gorillas infected with COVID-19 was found in a California zoo [53]. In rodents, HCoV-HKU1 is currently considered to be a rodent-related virus, originally obtained from infected mice (Fig 2) [54]. In 31 animals sampled on January 5, 2004, before culling of wild animals at a Guangzhou live animal market, including 4 cats, 3 red foxes and one Lesser rice field rat were tested SARS-CoV positive based on RT-PCR test [55].

Many natural hosts are classified under Laurasiatheria (Fig 2). Classified under the Rhinolophus genus of the order Chiroptera, bats are the host of SARS-CoV, MERS-CoV, HCoV-229E, and HCoV-NL63 based on the isolation of these viruses from bats and genomic sequencing confirmation (Fig 2 and Table 1). SARS-CoV is known to exist in greater horseshoe bats and Chinese rufous horseshoe bats (Table 1). The SARS-CoV-2 genome shows high homology to SARS-related coronaviruses identified in horseshoe bats [56]. The sequence homology between SARS-CoV-2 and SARS-CoV is 79.6% [56]. RaTG13, a bat coronavirus that shares 96% genetic similarity with SARS-CoV-2, was isolated from horseshoe bats [56]. However, the exact SARS-CoV-2 virus or its genome sequence has not been isolated from bats through natural infections.

Malayan pangolin (*Manis javanica*) is a species under Laurasiatheria. Viral metagenomics showed that several Malayan Pangolins host a variety of coronaviruses, among which SARS-CoV was the most widespread one [57]. Zhang et al. found a SARS-CoV-2-like CoV (named Pangolin-CoV) in dead Malayan pangolins, Pangolin-CoV is 91.02% and 90.55% identical to SARS-CoV-2 and BatCoV RaTG13, and concluded that except for RaTG13, Pangolin-CoV is the most closely related coronavirus to SARS-CoV-2 [58]. On March 26, 2020, Yi Guan [59] detected several coronaviruses in a small number of pangolins smuggled into China that are closely related to SARS-CoV-2. This similarity suggests that Malayan pangolin is likely an intermediate host directly involved in the current SARS-CoV-2 outbreak.

**Table 1. Verified natural hosts of human coronaviruses (in alphabetic order).**

| Host | Taxon ID | Human Coronaviruses | Evidence | References (PMID or reference citation) |
|---|---|---|---|---|
| Human | 9606 | SC, SC2, MC, OC43, HKU1, 229E, NL63 | Virus isolation, RT-PCR, serum antibody detection | 15347429, 26695637, 29551135, 32798769 |
| Alpaca | 9655 | MC | Serum antibody detection | 30832356 |
| Asian small-clawed otter | 452597 | SC2 | PCR | [45] |
| Bactrian camel | 9837 | MC | Antibody detection, virus isolation | 31969478 |
| Bat (e.g., Chinese rufous horseshoe bat) | 9397 (e.g., 89399) | SC, MC, 229E, NL63 | Genome sequencing | 30844511, 16195424, 30531947, 26695637, 29551135 |
| Binturong | 94180 | SC2 | PCR and sequencing | 36121159 |
| Black-Tailed Marmoset | 1090896 | SC2 | Viral RNA detection, immunohistochemistry detection, | 35577455 |
| Canada Lynx | 61383 | SC2 | RT-PCR | 36121159 |
| Cattle | 9913 | OC43 | Genome sequencing | 32629960,15650185 |
| Chinese ferret-badger | 204267 | SC | Neutralizing antibody detection, RNA isolation, RT-PCR | 12958366 |
| Civet | 9673 | SC | PCR, virus isolation, neutralizing antibody detection | 30844511 |
| Coatimundi | 743424 | SC2 | PCR and sequencing | [46] |
| Cougar | 9696 | SC2 | PT-PCR | 36121159, 35062324 |
| Dog | 9615 | SC2 | Genome sequencing, virus isolation | 32408337 |
| Domestic cat | 9685 | SC, SC2 | ELISA, neutralization assay RT-PCR | 15921605, 32402157 |
| Domestic mouse | 10090 | HKU1 | PCR | 19239338 |
| Dromedary camel | 9838 | MC, 229E | Antibody detection, virus isolation | 31969478, 24896817, 27528677 |
| Fishing cat | 61388 | SC2 | PCR and genome sequencing | 36121159 |
| Giant anteater | 71006 | SC2 | Viral RNA detection | 36692797 |
| Gorilla | 9593 | SC2 | Viral RNA detection | [47] |
| Hippopotamus | 9833 | SC2 | PCR | [48] |
| Indian Leopard | 421001 | SC2 | Genome sequence | 36121159 |
| Large hairy armadillo | 29080 | SC2 | RT-PCR | [49] |
| Lesser rice-field rat | 69075 | SC | RT-PCR | 15921605 |
| Lion | 9689 | SC2 | Virus detection | 33051368 |
| Malayan pangolin | 9974 | SC, SC2 | Metagenomics prediction | 31652964, 32197085 |
| Masked palm civet | 9675 | SC | PCR, virus isolation, neutralizing antibody detection | 12958366, 23671097 |
| Mink | 9655 | SC2 | Genome sequencing, PCR | 33172935, 32663073 |
| Pangolin | 9974 | SC2 | Genome sequencing | 32724171 |
| Raccoon dog | 34880 | SC | RT-PCR, virus isolation, neutralizing antibody detection | 12958366 |
| Red fox | 9627 | SC | RT-PCR | 16485471 |
| Snow Leopard | 29064 | SC2 | PCR | 36121159 |
| Spotted hyenas | 9678 | SC2 | PCR | 36121159 |
| Tiger | 9694 | SC2 | Virus detection | 33051368 |
| Virginia opossum | 9267 | SC2 | PCR | [50] |
| White-footed mouse | 10041 | SC2 | RT-PCR, and serum antibody detection. | [50] |
| White-tail deer | 9874 | SC2 | rPT-PCR | 34942632, 3578920 |

Abbreviations: SARS-CoV-2: SC2; SARS-CoV: SC; MERS-CoV: MC; HCoV-OC43: OC43; HCoV-HKU1: HKU1; HCoV-229E:229E; HCoV-NL63: NL63.

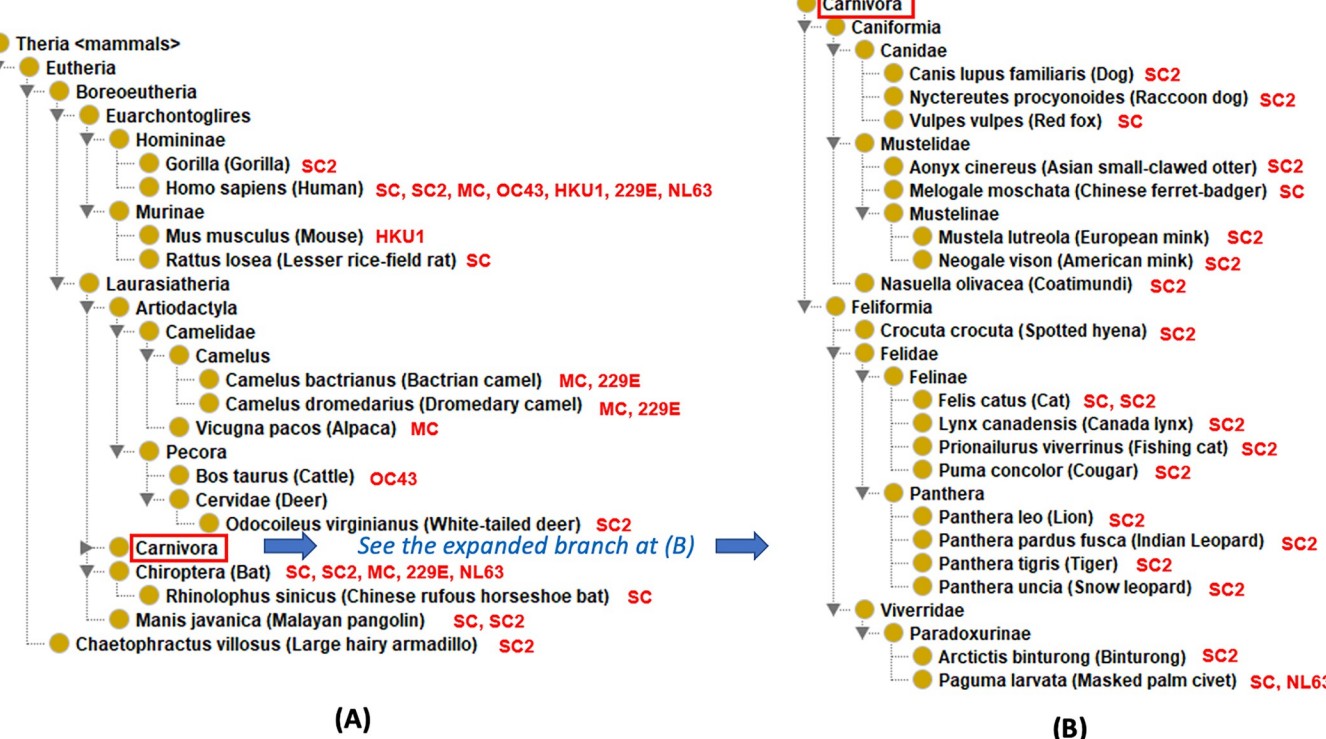

**Fig 2. Taxonomical hierarchy of 37 verified natural hosts of human coronaviruses.** All these hosts belong to Theria mammals. The hierarchy of the Carnivora animal hosts is singled out for optimal visualization. The added red-color name abbreviations indicate the coronaviruses capable of infecting the corresponding animal host. Abbreviations: SARS-CoV-2: SC2; SARS-CoV: SC; MERS-CoV: MC; HCoV-OC43: OC43; HCoV-HKU1: HKU1; HCoV-229E:229E; HCoV-NL63: NL63.

Cattle, camel, and alpacas are classified under the Artiodactyla order (Fig 2). Camels, including Bactrian camels, dromedaries, and hybrid camels, are important intermediate hosts for MERS-CoV [60]. MERS infection has also been found in alpacas that live with camels [61]. HCoV-229E was also isolated from alpacas raised in captivity with dromedary camels [6]. Using gene sequencing technology, Vijgen et al. found HCoV-OC43 and bovine coronavirus (BCoV) have remarkable antigenic and genetic similarities, and BCoV and HCoV-OC43 had a relatively recent zoonotic transmission event with their common ancestor likely dated to around 1890 [62].

The largest number of animal species found to host human coronaviruses are under the Carnivora order (Fig 2). Under the Canidae family, dogs, raccoon dogs, red foxes, Chinese ferret-badger, and mink were found to host SARS-CoV or SARS-CoV-2 (Fig 2 and Table 1). The raccoon dog was first identified in 2003 as an intermediate host of SARS in addition to civets [55]. Pet dogs were first reported to be infected with SARS-CoV-2 in Hong Kong and then reported subsequently in other places around the world [63]. SARS-CoV-2 was isolated from minks (under Mustelidae) on farms in the Netherlands, leading to mass culling [64]. Classified under the Feliformia suborder, domestic cats, lions, tigers, and civets were also found to be infected with human coronaviruses (Fig 2 and Table 1).

## Identification and classification of 19 verified laboratory animal hosts of human coronaviruses

Laboratory animal models have been widely explored to study human coronaviruses with host-virus interaction mechanisms and translational drug/vaccine studies. Our literature

**Table 2. Laboratory models of human coronavirus hosts (in alphabetic order).**

| Host | Taxon ID | Human Coronaviruses | Evidence | References (PMID) |
|---|---|---|---|---|
| African green monkey | 9534 | SC | Moderate to high titers of SARS-CoV | 15527829 |
| Alpaca | 30538 | MC | Nasal swab specimens, serum samples | 27070385 |
| Baboon | 9557 | SC2 | Detect viral RNA in nasopharyngeal swabs | 33340034 |
| Bank vole | 447135 | SC2 | qRT-PCR | 33754987 |
| Civet | 9673 | SC | RT-PCR | 17037579 |
| Common Tree Shrew | 9395 | SC2 | Virus detection | 32994418 |
| Common marmoset | 9483 | SC | RT-PCR | 16049331 |
| | | MC | qRT-PCR | 25144235 |
| Cynomolgus macaque | 9541/ 36519 | SC | Nasal, oral and rectal swabs, RT-PCR | 12748632, 21533129 |
| | | MC | Antibodies detection in sera, RT-qPCR | 32303590 |
| | | SC2 | Antibodies detection in sera, RT-qPCR | 32303590 |
| Ferret | 9669 | SC | Infected intranasally with 10(3) TCID50 SARS-CoV | 18234270 |
| | | SC2 | Viral detection | 32269068 |
| Fruit bat | 9407 | SC2 | RT-qPCR, antibody detection | 32838346 |
| Mouse | 10090 | SC | Prior infection, neutralizing antibody detection | 15016880 |
| | | MC | Serum neutralizing antibodies and MERS-CoV S1 protein-specific IgG antibodies detection, even death | 26446606 |
| | | SC2 | expressing hACE2 receptors, PT-PCR | 32380511, 36222118 |
| North American raccoon | 9654 | SC2 | Intranasally inoculated, rPT-PCR, seroconverted | 35097038 |
| North American deer mouse | 10042 | SC2 | Virus detection in nasal washes, oropharyngeal and rectal swabs | 34127676 |
| Raccoon dog | 34880 | SC2 | Viral replication and tissue lesion in nasal conchae | 33089771 |
| Rhesus monkey | 9544 | SC | RT-PCR, virus isolation | 15892035 |
| | | MC | qRT-PCR | 24218506 |
| | | SC2 | Virus detection | 32396922 |
| Rabbit | 9986 | SC2 | qRT-PCR, viral RNA positive in the nose and throat for at least four days | 33356979 |
| Sheep | 9940 | SC2 | RT-qPCR of respiratory tract tissues and lymphoid tissues | 34816258 |
| Syrian hamster | 10036 | SC | Virus detection | 17037579, 17499378 |
| | | SC2 | Virus detection | 32215622, 36222118 |
| Striped skunk | 30548 | SC2 | rPT-PCR | 35091038 |

**Abbreviations:** SARS-CoV-2: SC2; SARS-CoV: SC; MERS-CoV: MC.

mining identified 19 laboratory animal models that have been used in various laboratory studies on human coronavirus hosts (Table 2). Human coronaviruses have been detected in these laboratory animals using experimental methods from different anatomical locations such as saliva, blood, and lungs, and many of the infected animals can develop symptoms and replicate in vivo (Table 2).

All 19 laboratory animal models belong to Boreoeutheria, a clade (magnorder) of Eutheria (i.e., placental mammals), which is also under therian mammals (Fig 3). More specifically, these laboratory animals are categorized under two clades of Boreoeutheria: Euarrchontoglires and Laurasiatheria (Fig 3). Different from the natural hosts, the list of existing laboratory animals does not include any species under the superorder Xenarthra.

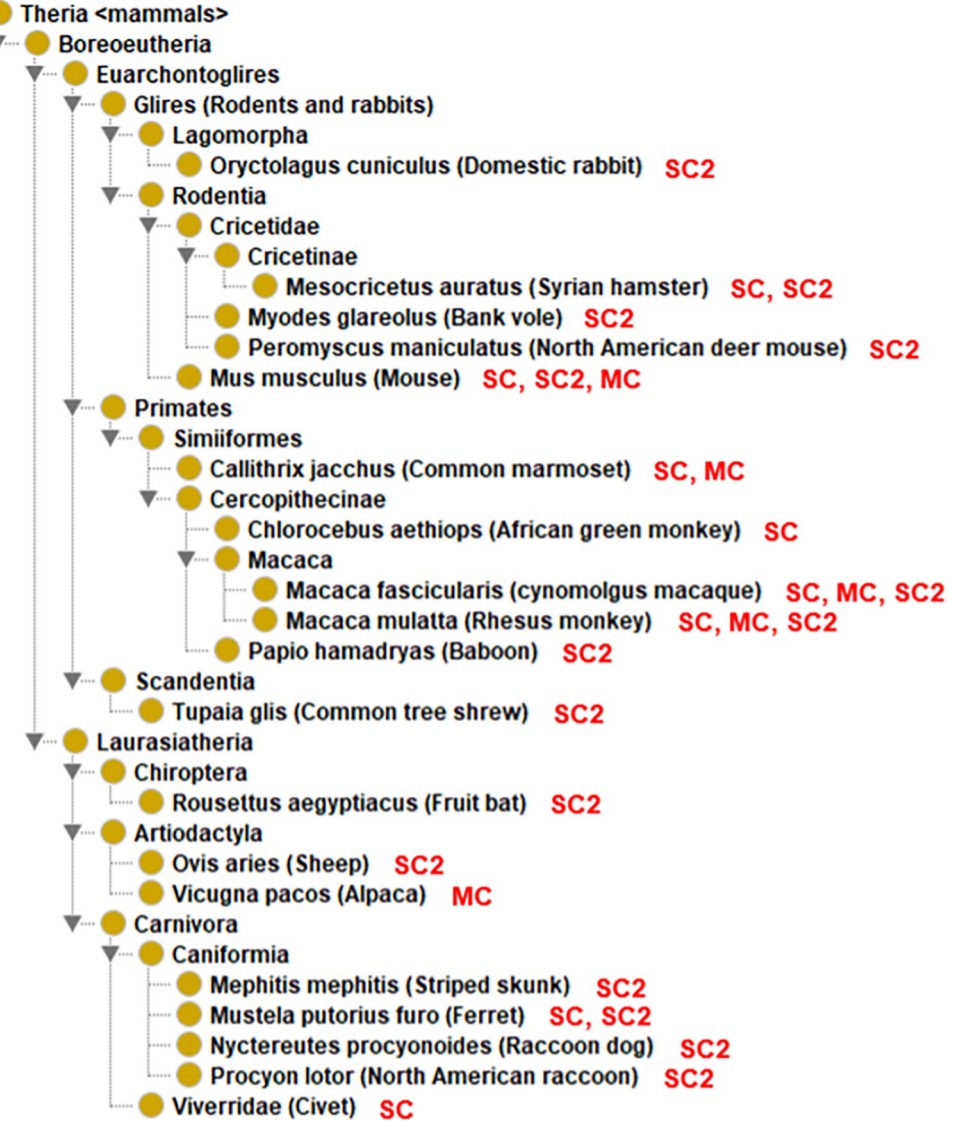

**Fig 3. Taxonomical hierarchy of 19 laboratory animal hosts of human coronaviruses.** All these hosts belong to Boreoeutheria, a special group of mammals. Abbreviations: SARS-CoV-2: SC2; SARS-CoV: SC; MERS-CoV: MC.

Ten laboratory animals are categorized under four orders Primates, Rodentia, Lagomorpha, and Scandentia within the Euarrchontoglires clade. Primates are the order of animals phylogenetically close to humans and ideal laboratory animal models for human coronaviruses [65–67]. These laboratory animals under Primates are all Simiiformes (infraorder), including marmosets, African green monkeys, and macaques. These non-human primates are frequently used for COVID-19 research; however, the usage of these animals is also expensive [68].

The orders Rodentia and Lagomorpha belong to the clade Glires. Under Glires are four laboratory animal species, including the Syrian hamster, bank vole, deer mouse, and domestic mouse, and domestic rabbits (Fig 3). The first four belong to Rodentia and the last one belongs to Lagomorpha. The Lagomorpha have a second pair of incisor teeth behind the first pair in the upper jaw, while Rodentia has only one pair. The mouse is the most commonly used animal model for coronavirus research [69,70]. Mouse models are popular because of their

affordability, availability, and clear genetic backgrounds, and they have been widely used for studying the pathogenesis of human coronaviruses [71]. Mouse is a natural host for human coronavirus strain HKU1 (Fig 2) [72]. However, mice infected naturally with SARS-CoV-2 show fewer symptoms and low virus replication [73]. To make mice a better laboratory model for coronavirus studies, genetically modified mice are often used as detailed in the next section.

Under the Laurasiatheria clade are seven animals separated into two orders: Artiodactyla and Carnivora. Sheep and alpaca are under Artiodactyla. The five species under Carnivora include striped skunks, domestic ferrets, raccoon dogs, and North American Raccoon (Fig 3). Compared to the large number of Laurasiatheria animals in the group of natural human coronavirus hosts, the number of Laurasiatheria animals in the laboratory host group is relatively smaller, likely due to their expensive cost and fewer related reagents available for deep investigations.

## Mouse models developed for human coronavirus studies

To further improve the laboratory mouse model for coronavirus studies, continuous virus passaging of coronaviruses in wild-type and even genetically modified mice was frequently used to generate mouse-adapted coronaviruses [17,74]. For example, a clinical isolate of the SARS-CoV-2 strain was serially passaged for 6 generations in the respiratory tract of aged BALB/c mice, resulting in the generation of a more infectious genetically modified strain called MASCp6 [75]. Adaptive mutations, including an N501Y mutation in the spike protein receptor binding domain, were later identified by deep sequencing of the MASCp6 genome [75]. Similarly, Roberts et al. [74] generated the mouse-adapted SARS-CoV-2 MA15 strain after 15 passages of SARS-CoV-2 in BALB/c mice, and the resulting MA15 became lethal for mice following intranasal inoculation. Li and McCray utilized human DPP4 knock-in (hDPP4 KI) mice to infect MERS-CoV, but the transgenic mice still did not display respiratory disease after MERS-CoV infection. After serial passages of 30 generations in vivo, the wild-type MERS-CoV strain eventually became $MERS_{MA}6.1.2$, which produced significantly higher titers than the parental virus strain in the lungs of hDPP4 KI mice and caused diffuse lung injury and a fatal respiratory infection [76].

To increase the infection rate, different transgenic mice were developed and utilized [69]. A comparative study showed that the genetically modified mouse model is better than the wild-type mouse model with adenovirus-delivered hACE2 [73], suggesting that genetically modified mouse models are the preferred model for coronavirus studies. The commonly used genetically modified mouse models for SARS-CoV, MERS-CoV and SARS-CoV-2 were collected and provided in Table 3.

Typically, transgenic mouse models were typically generated to express hACE2 (human angiotensin-converting enzyme 2) or hDPP4 (dipeptidyl peptidase-4). ACE2 is the host receptor that binds the S protein in SARS-CoV and SARS-CoV-2, and the DPP4 is the receptor binding the S protein in MERS-CoV [77]. An example of the genetically modified mouse model is the HFH4-hACE2 C3B6 mouse that expresses human ACE2 under the control of a lung-ciliated epithelial cell-specific HFH4/ FOXJ1 promoter [78,79]. HFH4-hACE2 mice expressed high levels of hACE2 in the lung but at varying expression levels in other tissues, including the brain, liver, kidney, and gastrointestinal tract [80].

Methods for generating hACE2 or hDPP4 mice differ (Table 3). Different promoters are used to drive the expression of the hACE2 or hDPP4 gene in mice. For example, the most commonly used mouse models of SARS-CoV and SARS-CoV-2 are transgenic mice with human cytokeratin 18 as the promoter and human ACE2 added [68]. In addition, the chicken-β actin

**Table 3. Genetically modified mouse models for SARS-CoV, MERS-CoV, SARS-CoV-2.**

| No. | Transgenic mouse model | Features | Virus | Symptoms | PMID |
|---|---|---|---|---|---|
| 1 | K18-hACE2 transgenic (Tg) mouse | Expressing hACE2 under human keratin 18 promoter | SARS-CoV | weight loss, lethargic, labored breathing | 17079315 |
| | | | SARS-CoV-2 | weight loss | 32723427 |
| 2 | hACE2 transgenic mice (AC70) | Expressing hACE2 under chicken β-actin promoter (CAG) | SARS-CoV | weight loss | 17108019 |
| 3 | Human DPP4 knock-in (hDPP4 KI) mice | hDPP4 knock-in replacing mDPP4 | MERS-CoV | weight loss | 31883095 |
| 4 | hDPP4 transgenic mice | Expressing hDPP4 with cytokeratin 18 promoter | MERS-CoV | weight loss, hypothermia | 26486634 |
| 5 | CAG-hACE2 mice | C57BL/6J mice expressing hACE2 under CAG promoter | SARS-CoV | lethal pneumonia & CNS inflammation | 17108019 |
| | | | SARS-CoV-2 | pulmonary injury, Most mice died | 34463644 |
| 6 | HFH4-hACE2 transgenic mice | Expressing hACE2 under HFH4 (FoxJ1) promoter | SARS-CoV | weight loss and death | 26976607 |
| | | | SARS-CoV-2 | weight loss, 60% survived | 32516571 |
| 7 | hACE2 transgenic mice | Expressing hACE2 under control of mACE2 promoter | SARS-CoV-2 | slight bristled fur, weight loss | 32380511 |
| 8 | B6 K18-hACE2 mice | Expressing hACE2 under human cytokeratin 18 (K18) promoter | SARS-CoV-2 | weight loss, lethargy, ruffled fur, laboured breathing | 33073694 |
| 9 | A mouse with humanized ACE2 | Expressing hACE2 cDNA from endogenous mACE2 locus | SARS-CoV-2 | weight loss | 32485164 |
| 10 | Adeno-associated virus-hACE2 infected mouse | adeno-associated virus (AAV) | SARS-CoV-2 | weight loss, productive infection | 32750141 |
| 11 | AdV-hACE2-Transduced Mice | adenoviral vector | SARS-CoV-2 | 10% maximum weight loss | 32553273 |

promoter (for SARS-CoV infection) and HFH4/ Foxji promoter (for SARS-CoV-2 infection) were also used to generate transgenic mice. The CRISPR/Cas9 knock-in technology has also been used to generate a mouse model expressing human ACE2 (hACE2) [81].

Generally, genetically modified mice show significantly higher coronavirus infection rates and more severe symptoms than wild-type mice. For example, while wild-type C57BL/6 mice showed no or low viral loads after intranasal infection with SARS-CoV-2, young and aged genetically modified hACE2 mice sustained high viral loads in the lung, trachea, and brain [81]. Although SARS-CoV-2 infected-aged hACE2 mice survived, interstitial pneumonia and elevated cytokines were observed. It was also found that intragastric inoculation of SARS-CoV-2 caused viral infection and pulmonary pathological changes in hACE2 mice [81].

## Ontological and computational analysis of huanan seafood wholesale market animals

To trace possible COVID-19 host origin, we systematically analyzed the WHO report [30] of surveillance and analysis of the animals identified at the Huanan Seafood Wholesale Market at the beginning of the COVID-19 outbreak. A total of 40 animals were reported in the seafood market. Among these 40 animals, 30 species are mammals, and 9 species are Sauria, and one species is Salamanders under Andrias as shown in our ontology-based taxonomical analysis (Fig 4). The Sauria group includes Saltwater crocodiles, Siamese crocodiles, pigeon, goose, duck, turkey, chicken, Ring-necked pheasant, snake. Among these 30 mammals, 24 species are known natural or laboratory COVID-19 hosts (see Tables 1 and 2), and 6 species (i.e., Hystrix hodgsoni, goat, pig, puma, Spiny hedgehogs, Leschenault's rousette) have not been found to be infected with SARS-CoV-2. Seven animal species (e.g., dog, cat, ferret, rabbit, Malayan pangolin, raccoon dog, and bat) were found to be infectable with SARS-CoV-2 at the natural conditions. Nineteen species (e.g., Syrian hamster, common tree shrew, mouse, common marmoset,

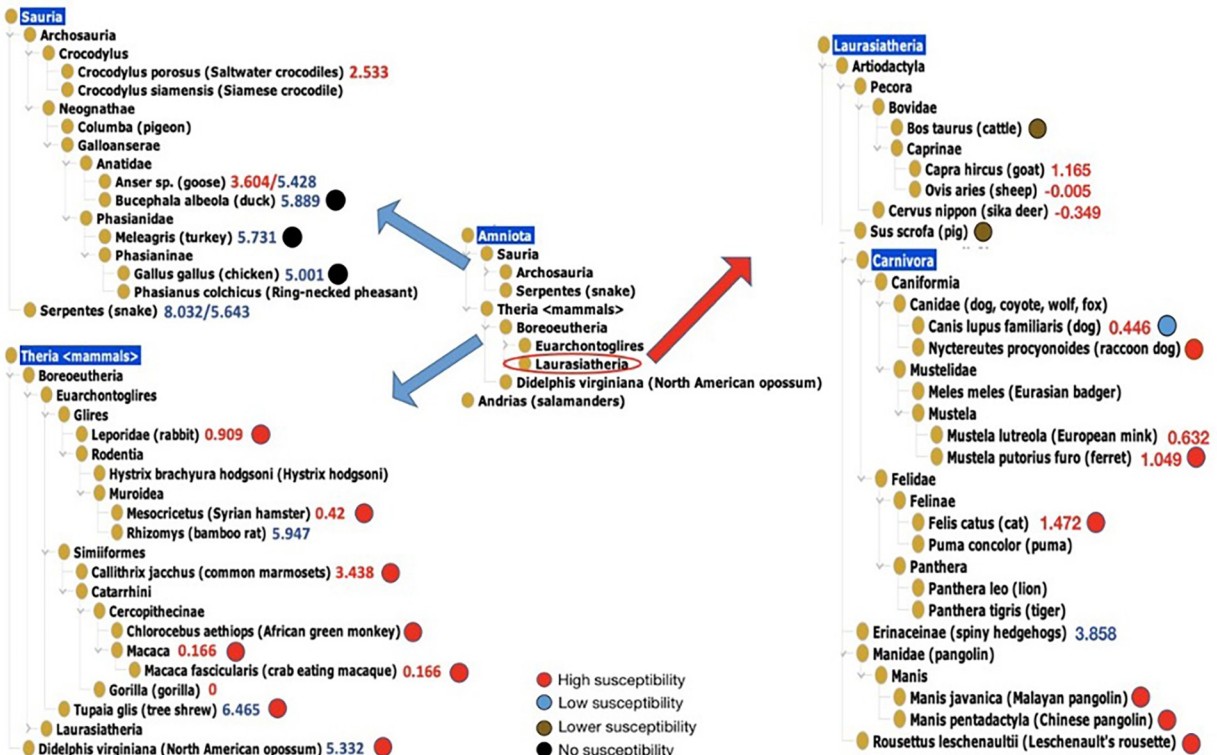

**Fig 4. Taxonomical classification and analysis of 40 animals found in Hunan Seafood Wholesale Market.** Four levels of susceptibility are labeled with colored circles based on the WHO report [30]. The added ΔΔG values were obtained from the study by Lam, et al. [31].

and Cynomolgus macaque) were found to be infectable with SARS-CoV-2 at the experimental conditions. Eighteen species (e.g., porcupine, sika deer, Crocodylus siamensis, and megalobatrachus) belong to wildlife animal species.

The WHO report also includes the susceptibility values of these animals, which helps our further analysis of their likelihood to be COVID-19 hosts. To further assess the susceptibility of being COVID-19 hosts, we compared the WHO susceptibility degrees with the calculated ability of the SARS-CoV-2 S-protein: ACE2 complex binding for different animal species. The SARS-CoV-2 S-protein is a viral adhesin critical to the invasion to the host cells. S-protein binds to the host angiotensin converting enzyme 2 (ACE2) [82], which initiates the viral infection process. Therefore, the capability of the S-protein-ACE2 binding in a specific animal provides an indication of the animal being potential SARS-CoV-2 host. Such binding capability was measured using the calculated changes in energy (ΔΔG) of the SARS-CoV-2 S-protein: ACE2 complex binding [31]. Lower ΔΔG values represent more stable binding, and therefore higher risk of infection. Our comparable study found that in general the WHO reported animal susceptibilities align well with the ΔΔG values. The animals with higher laboratory susceptibility, such as Ferret, Cat, Rabbit, Syrian Hamster, Crab Eating Macaque, had lower ΔΔG levels. The WHO report clearly indicates that dogs have low susceptibility as laboratory animals for SARS-CoV-2. However, it is worth noting that the ΔΔG value of dogs is lower, which slightly deviates from the previous result. Additionally, cattle and pigs exhibit lower susceptibility to SARS-CoV-2 infection. Except that the ΔΔG value of Tree Shrew and North American opossum was higher than 3.7, the results of other susceptible laboratory animals were consistent with the results of the calculation model. The ΔΔG value in duck, Turkey and chicken was

all greater than 5.0, suggesting that these animals are not able to be infected with SARS-CoV-2 (Fig 4).

## ACE2 phylogenetic analysis predicting coronavirus hosts

Since ACE2 is critical to the coronaviral binding to host cells, an ACE2-based phylogenetic analysis has been used to compare coronavirus binding capabilities among different coronavirus hosts and predict possible coronavirus hosts [83]. To confirm and further evaluate the potential value of the ACE2-based phylogenetic tree analysis for coronavirus host investigation, we performed a phylogenetic analysis using the whole sequences of ACE2 proteins in 49 animals, including snake, chicken, known human coronavirus hosts collected by our own meta-analysis (Tables 1 and 2), and other mammal species identified in Huanan Seafood Wholesale Market as described above (Fig 4). These ACE2 sequences from different host species of human coronaviruses were discovered in the NCBI Protein Database. Our preference is to select the mature form from the corresponding species, followed by isoforms, with the precursor of the protein sequence being the last option.

One significant finding from our study is that the ACE2 proteins of Western terrestrial garter snake and Chicken were phylogenetically far from those other mammals (Fig 5), which aligns with our hypothesis that snake and chicken were not susceptible to SARS-CoV-2 infection. The phylogenetic tree result is also aligned with the experimental observation that poultry was insusceptible to SARS-CoV-2 infection [84]. In general, all the known mammal hosts are positioned with its own branch of mammals in the ACE2-based phylogenetic tree structure, confirming the critical role of ACE2 as an indicator of being a SARS-CoV-2 infection host. Note that not all mammals in the phylogenetic tree are equally susceptible to human coronavirus infection. For example, mouse is less than susceptible compared to many other mammals (e.g., humans). However, genetically modified human ACE2 mice are more susceptible to COVID-19 than the wild type. Furthermore, some mammals (e.g., goat, pig, puma, and badger) identified in the Huanan Seafood Wholesale Market have not been found to be infected by SARS-CoV-2. The potential of these mammals as potential SARS-CoV-2 intermediate hosts deserves further investigation.

## Ontological modeling, query, and analysis of human coronavirus hosts

To support standardized and digitized modeling and representation of human coronavirus hosts, we represented our collected human coronavirus hosts in the Coronavirus Infectious Disease Ontology [16,37,85]. To represent the relations between coronavirus and its host, in CIDO we generated a new ontology relation called '*capable of infecting host*', which represents a relation between a pathogen and a host in which the pathogen has verified evidence of infecting the host. For example, the following axiom relation defined in CIDO represents that SARS-CoV-2 is capable of infecting white-tail deer (which has the scientific name '*Odocoileus virginianus*' (Fig 6):

SARS-CoV-2: 'capable of infecting host' some 'Odocoileus virginianus'

On the other side, CIDO uses the following axiom to represent the role of white-tail deer as a COVID-19 host (Fig 6):

'white-tail deer': 'has role' some 'SARS-CoV-2 natural host role'

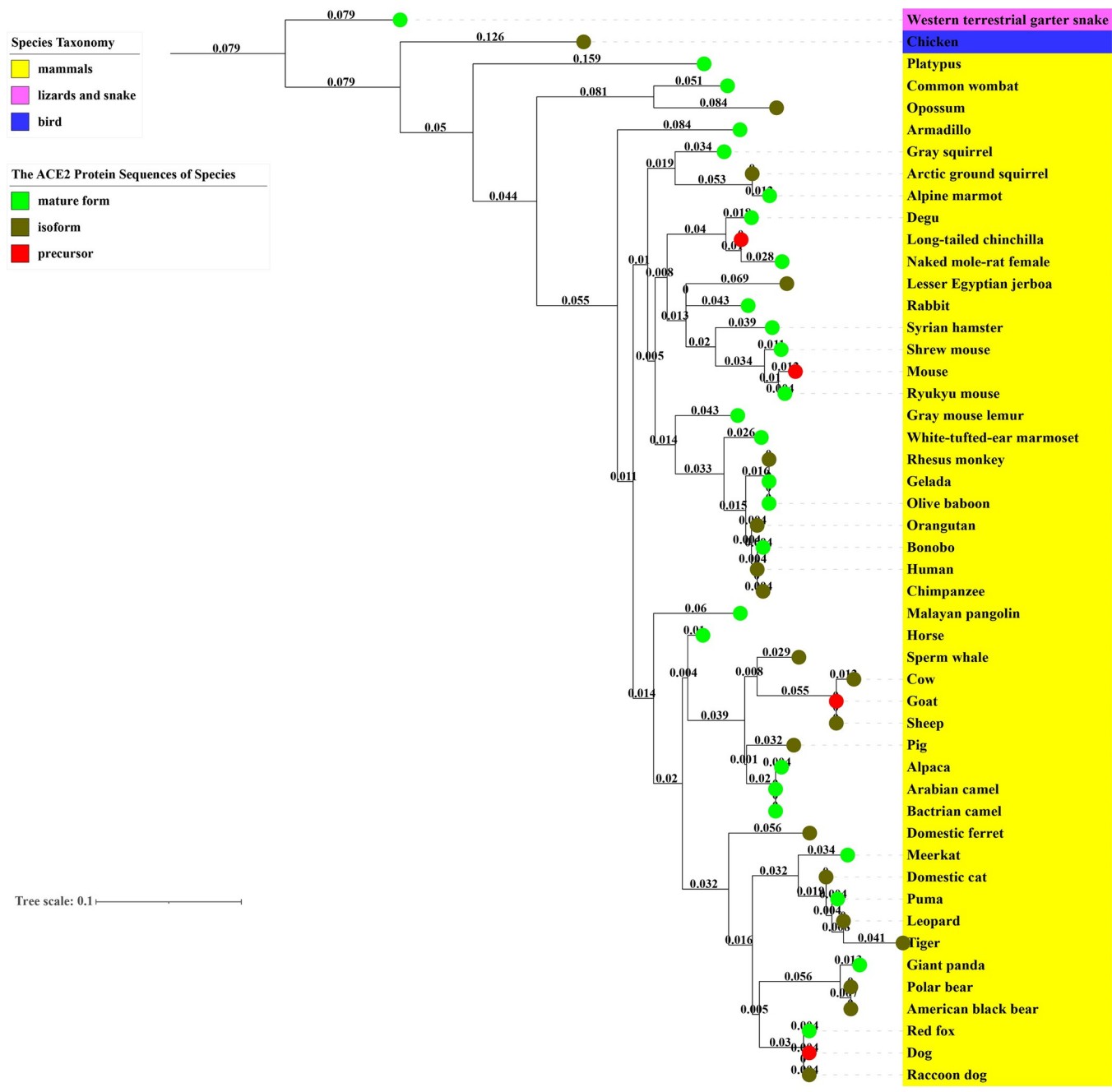

**Fig 5. Phylogenetic analysis of ACE2 proteins from 49 animal species.** The phylogenetic tree is drawn to scale with branch lengths measured in the number of substitutions per site. The scale bar indicates nucleotide substitutions per site.

Using such a strategy, we have represented all the human coronaviruses and their infected hosts as identified in Tables 1 and 2 and S1 File.

For genetically modified mouse models (Table 3), since they are not represented in NCBI-Taxon ontology or NCBI Taxonomy database, we have generated new terms of these specific genetically modified mouse terms in CIDO and developed new logic axioms to represent their properties. For example, the CAG-hACE2 mouse model expresses the human ACE2 gene, which is ontologically represented as follows:

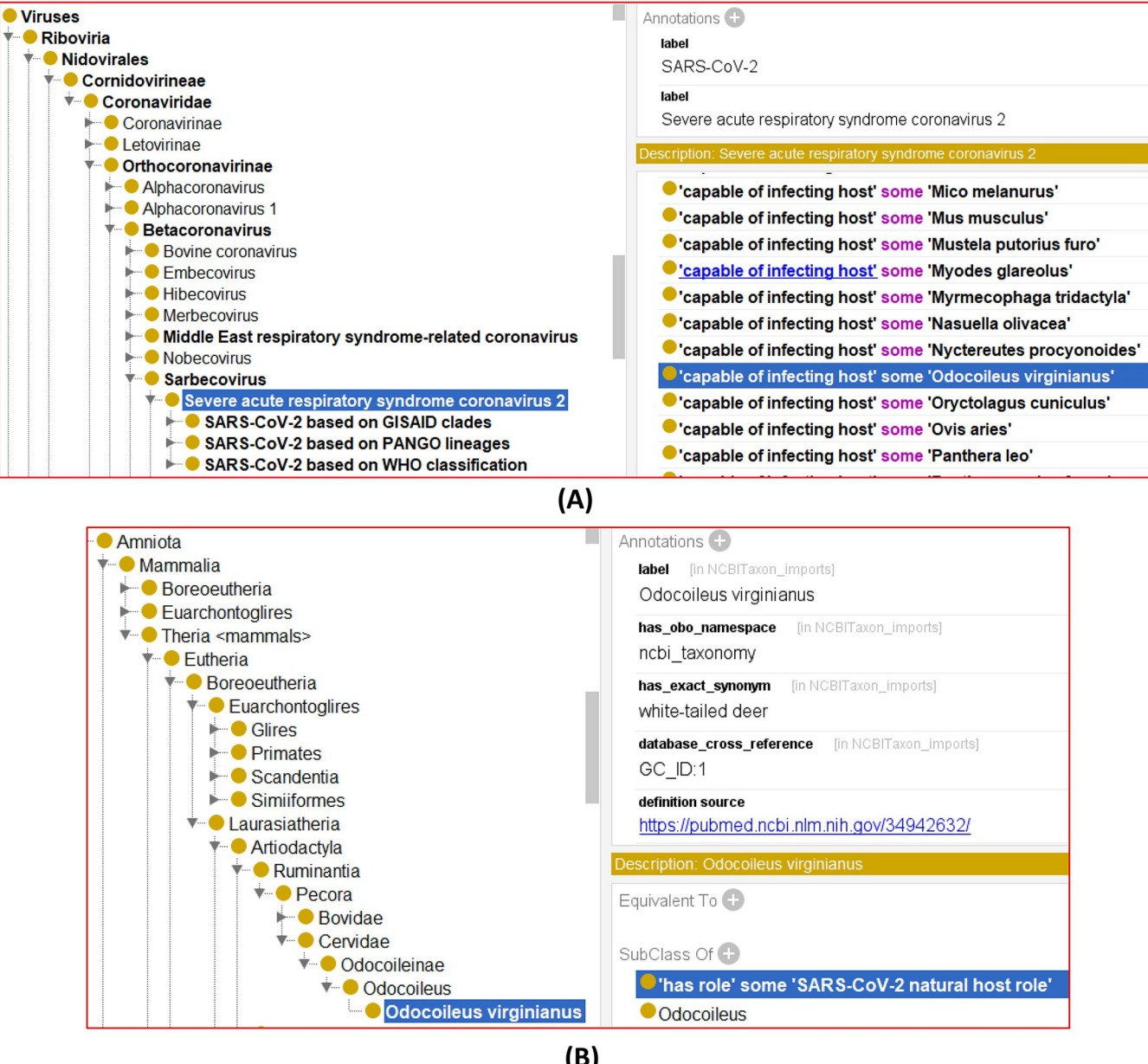

**Fig 6. CIDO representation of SARS-CoV-2 hosts.** (A) CIDO representation of SARS-CoV-2 in the hierarchical structure of viruses and its capability of infecting many hosts. (B) CIDO representation of white-tailed deer (i.e., *Odocoileus virginianus*) in the hierarchical structure of therian mammals and its role as SARS-CoV-2 natural host.

CAG-hACE2 mouse: expresses some 'angiotensin-converting enzyme 2 (human)'

To demonstrate the usage of our ontological modeling, we provide a SPARQL program to demonstrate its usage for advanced queries. Fig 7 provides a demonstration of a SPARQL script used to find the number of verified organisms that are capable of being infected by the SARS-CoV-2 virus through the SPARQL query of knowledge stored in CIDO. Specifically, 38 hits were identified from the SPARQL script execution. S2 File provides another SPARQL query that lays out the specific details of these 38 SARS-CoV-2 hosts.

```
# Goal: identify the total number of SARS-CoV-2 hosts

PREFIX capable_of_infecting_host: <http://purl.obolibrary.org/obo/CIDO_0001195>
PREFIX SARS_CoV_2: <http://purl.obolibrary.org/obo/NCBITaxon_2697049>
SELECT count(distinct ?x)
from <http://purl.obolibrary.org/obo/merged/CIDO>
WHERE {
        ?x rdf:type owl:Class .
        ?x rdfs:label ?label .
        SARS_CoV_2: rdfs:subClassOf ?restriction .
        ?restriction owl:onProperty capable_of_infecting_host:; owl:someValuesFrom ?x .
}
```

Output format [Table ▼] Max Rows [10 ▼]

[Run Query] [Reset]

[Result] [Raw Request/Permalinks] [Raw Response]

| callret-0 |
|-----------|
| 38 |

**Fig 7. SPARQL query for SARS-CoV-2 hosts stored in CIDO.** This SPARQL identified 37 organisms that are capable of being infected by the SARS-CoV-2 virus. The SPARQL was performed using the Ontobee SPARQL endpoint (https://ontobee.org/sparql). Another SPARQL query script that lists the names of these 38 hosts is provided in S2 File. Detailed information about these 38 hosts is also provided in S1 File, which offers the Excel sheets of the detailed hosts for different human coronaviruses.

## Discussion

The contributions of this article are multiple. Firstly, we systematically surveyed, identified, and collected 37 natural and 19 laboratory animal hosts of human coronaviruses (including SARS-CoV, SARS-CoV-2, MERS-CoV, and four less-virulent human coronaviruses) with experimental evidence. Secondly, our ontology-based taxonomical analysis found that all the verified natural and laboratory human coronavirus hosts are therian mammals, with the major category being Eutheria mammals (or placental mammals). The observation led us to hypothesize that human coronavirus hosts are therian mammals (see more discussion below). Thirdly, our mouse model meta-analysis identified 11 genetically engineered mouse models that were typically developed to express humanized ACE2 or DPP4 in order to become more susceptible to SARS-CoV, MERS-CoV, and SARS-CoV-2 infection. Furthermore, a series of viral passages in mice were also often implemented to increase the pathogenesis/virulence of the coronaviruses in the mouse models. Fourthly, we have modeled and represented these terms in the CIDO ontology and demonstrated their usage using SPARQL. Finally, based on the findings from this study, we propose a dynamic host-coronavirus interaction MOVIE model as described below.

Our ontological methods are two-fold. Firstly, our taxonomical analysis is an ontology based. Specifically, the whole taxonomical analysis performed in our manuscript was based on NCBI Taxonomy Ontology (i.e., NCBITaxon). After we mapped the animal types to specific ontological identifiers in NCBITaxon, we used the tool Ontofox to extract these animal names and the names of their specific ancestors, and then used the Protégé-OWL editor to build up the taxonomical trees as seen in the specific taxonomical figures. Secondly, we used the CIDO for modeling and representation of verified knowledge related to human coronavirus hosts.

The extracted NCBITaxon ontological results were added to the CIDO ontology and become parts of the CIDO ontology. Furthermore, the CIDO ontology is extended to include other information such as transgenic mice and their transgenic modification and susceptibility to coronavirus infection. The incorporation of the known information in the CIDO make it feasible to develop advanced applications such as the SPARQL query, data integration and standardization, machine learning, natural language processing[18,45,86].

Our taxonomical classification provides a way to identify patterns in all the experimentally verified natural or laboratory animal hosts of human coronaviruses, leading to our proposal of the "Therian Host Hypothesis", i.e., human coronavirus hosts are all therian animals. The therian host hypothesis of human coronaviruses accords with what we have known about the taxonomical classification of animal hosts of human coronaviruses. Being a scientific hypothesis, it may be proven incorrect if later contradicted observations are found. However, it is worth raising this hypothesis for now, since this hypothesis can be quite useful and can be used to infer that non-therian mammals are not human coronavirus hosts. For example, according to the therian host hypothesis, since the snake is a reptile animal under the Sauropsida clade instead of a therian mammal, we then infer that the snake is not a host for human coronaviruses. Overall, the therian host hypothesis helps us to narrow the range of the possible extent of natural hosts, and carefully select laboratory animal hosts for experiments, and find an appropriate surveillance approach for the transmission of the zoonotic disease.

Controversial data have been generated in terms of the status of the snake as a host of human coronaviruses as there has not been any detected snake that is a host of coronaviruses. A computational bioinformatics study predicted the snake to be a coronavirus potential carrier host based on its similar genetic codon usage bias with 2019-nCoV or through evolutionary analysis (e.g., SARS-CoV-2) [87,88]. However, a new study focusing on the analysis of the interactions between the receptor-binding domain (RBD) of the SARS-CoV-2 S protein and the 20 key amino acid residues in the receptor ACE2 proteins from a list of mammals, birds, turtles, and snakes showed different results [89]. Specifically, nearly half of the key residues in snakes and turtles were changed. A structure simulation study showed that when a contact amino acid (AA) in hACE2 is changed to a smaller AA in a snake, the contact force for the protein-protein interaction will be reduced [89]. This study concluded that snakes (and turtles) are not intermediate hosts for SARS-CoV-2.

Our therian animal hypothesis also excludes reptiles, birds, and insects being human coronavirus hosts. Theria is a clade under the Mammalia class. Reptiles such as snakes and turtles, birds such as chickens and eagles, and insects such as houseflies and mosquitoes do not belong to Mammalia or Theria. Therefore, according to our hypothesis, we can exclude these non-therian animals to be human coronavirus hosts. Taxonomically, there are four non-therian orders of mammals: the living Monotremata, the extinct Tricono-donta, Docodonta, and Multituberculata. Examples of living Monotremata include Platypus and long-beaked echidnas, which are all distributed in Oceania. These living Monotremata lay eggs; but like all mammals, the female monotremes nurse their babies with milk. Based on our hypothesis, these living Monotremata mammals are not potential human coronavirus hosts. However, since limited data is available, more studies are required to assess our conclusions.

To further address the mechanism of the COVID-19 host origination and transmission, we propose a new MOVIE model—"Multiple-Organism viral Variations and Immune Evasion" based on the findings reported in this study and from the literature. The MOVIE model states that the complex dynamic interactions between COVID-19 viruses and hosts occurs in multiple host organisms in which the viruses undergo continuous genetic variations and immune evasion under various viral, host, and environmental conditions. The viral variation is the

major mechanistic process, through which immune evasion occurs to achieve survival in the host. For the COVID-19 pandemic, our MOVIE model has the following tenets:

(1) **SARS-CoV-2 Origination:** SARS-CoV-2 originated from continuous viral genetic mutations/variations in one or more host species that led to increasing viral virulence in the species. As shown in our annotated laboratory animal models (Table 3), the viral virulence turned to steadily increase in the same laboratory animal model in the early generations of viral passages in the host species. More evidence includes the early virulence increasing of SARS-CoV-2 variants including Alpha, Beta, and Delta strains. The sufficient genetic mutations in one host species might make the virus more susceptible to infect another species such as humans. We can further hypothesize that SARS-CoV-2 had gone a long time of hidden genetic variations without human notice, virulence increasing, and transmission in different animal species before the human outbreak observed by humans.

(2) **SAR-CoV-2 Viral Immune Evasion:** SARS-CoV-2 viruses survive and evolve over time to achieve immune evasion in the hosts. Human coronaviruses conduct continuous genetic mutations and immune evasion under various host and environmental pressures. Numerous genetic variations in SARS-CoV-2 occurred during the coronavirus evolution from their continuous interactions with the hosts [90]. Such variations likely make the virus achieve "immune escape", i.e., escaping from the host immune system and then becoming more adaptive in the host [91,92].

(3) **Multi-host COVID-19 Infections:** Multiple COVID-19 hosts exist, and susceptible SARS-CoV-2 viral hosts are therian animals. The tenet is supported by our current findings that all experimentally verified natural, and laboratory human coronavirus hosts are under the Theria. Furthermore, the infection appears to be largely dependent on the SARS-CoV-2 S-protein:ACE2 complex binding capability, which can be estimated by the change in energy ($\Delta\Delta G$) of the binding. Specific coronavirus genetic mutations might have occurred in the Spike gene to make the virus more capable of infecting the hosts by S-protein:ACE2 binding.

(4) **Spiral HCI Dynamics:** The dynamic host-coronavirus interaction (HCI) is a complex system displayed as a spiral model in terms of its dynamic infection, transmission, and outcome manifestation. The "spiral" model of the complex HCI dynamics has at least two meanings: First, while the coronaviruses likely increase their virulence via genetic mutations to achieve immune evasion and host adaption at the early stage, the viral virulence likely decreases and the transmission likely increases over time due to intensive dynamic HCI processes at the population level. Correspondingly, the Delta variant is more pathogenic than the earlier variants, and the later Omicron variant is less pathogenic than the Delta variant. Second, while the first meaning provides a long-term effect, the short-time viral effects to the hosts are likely swing with unpredicted health outcomes.

The Spiral HCI Dynamics tenet is supported by the mathematical Susceptible-Exposed-Infectious-Removed (SEIR) model for infectious disease dynamics [93,94] and our proposed Spiral Symptom Occurrence Model Hypothesis [95]. Compared to the basic SIR model that has three groups (Susceptible, Infectious, and Recovered), the SEIR model adds the Exposed group for the latent period between being infected and becoming infectious. The SEIR model exhibits periodicity with open epidemic, and it identifies spiral waves over time that converge to the endemic equilibrium, and the model can also account for symptoms, transmission routes, and age [93]. The spiral symptom occurrence model hypothesis addresses the observed absence-presence-absence pattern of many special COVID-19 symptoms (e.g., loss of smell and taste) by spiral dynamics of host-coronavirus interactions through viral genetic variations,

host responses, and viral immune evasion under various conditions [95]. The coronaviruses might have first randomly mutated into more virulent and transmissible variants leading to more death and special symptoms; however, given human responses due to natural infection or vaccination, those viral variants with increasing transmission rate survive better, and these fast-transmitting variants tend to show less virulent manifestation. One application of the spiral HCI model is to identify certain genetic mutations that are responsible for specific disease manifestations (e.g., death and loss of smell and taste), which may be present in some variants (e.g., Alpha and Delta variants) but not in the others (e.g., Omicron strains).

Our proposed MOVIE model is aligned with and can be used to provide the mechanisms underlying many existing hypotheses. There have been three prevalent hypotheses regarding the evolutionary history of SARS-CoV-2 [96]. The first hypothesis states that COVID-19 viruses could have evolved in a "cryptically spread" way, which acts so that specific variant Omicron strains likely circulated in "covert transmission" among in a population(s) with insufficient viral surveillance and sequencing [97,98]. The second hypothesis is that the Omicron variant may have evolved in a patient with long-term infection with COVID-19, who might have a compromised immune system [97,98]. The single patient's condition would allow the virus to adapt in a long term in the host. Many studies have reported that many viral mutations do exist in severely immune-compromised patients including those with AIDS and cancer [99–101]. A third hypothesis is that the Omicron variants might have accumulated mutations in their nonhuman hosts and then transmitted into humans through infection [99–101]. Several studies strongly argued that Omicron mutations were acquired from non-human hosts [97,98]. Basically, the first "cryptically spread" hypothesis aligns with our SARS-CoV-2 Origination tenet that involves possible hidden genetic variations and transmission, which is accompanied with viral virulence increase at the early stage of outbreak. The second single-patient-incubation hypothesis can be explained by our viral immune evasion tenet that the single immunocompromised patient provides an ideal environment for the viral genetic variation and further immune evasion. The third nonhuman-to-human hypothesis can be explained by the Multi-host COVID-19 Infection tenet in our MOVIE model.

The MOVIE model incorporates the therian host hypothesis and predicts that SARS-CoV-2 originated from one or more therian host species; however, the model currently cannot inform exactly which species was the first host species. More research, including experimental analysis, is needed to further trace the origin of COVID-19 host species, which will support deep understanding of the disease etiology and transmission and guide rational prevention and control.

Ontological, theoretical, and computational methods can be used to deeply study the MOVIE model. Previously we proposed a set of four host-pathogen interaction (HPI) postulates and their corresponding ontological framework to study HPI such as host-coronavirus interaction [17]. These HPI postulates include the evolutionary dispositions involved in HPIs, the HPI dynamics, roles of HPI components leading to HPI outcomes, and HPI checkpoints critical for specific disease outcomes. Furthermore, we propose an HPI Postulate and Ontology (HPIPO) framework that uses interoperable ontologies for systematical modeling and representation of various granular details and knowledge within the scope of the HPI postulates, which would greatly support AI-ready data standardization, sharing, integration, and analysis. These postulates and framework can be used to further study the MOVIE model. Our further CIDO ontological representation makes our collection and annotations machine-interpretable in a way that computers can understand the information. Inside the CIDO ontology, such information is also seamlessly integrated with information such as protein-protein interactions and drug-target interactions. The systematical combination of all the information will allow us to better study the mechanisms of the interactions between different therian hosts and human

coronaviruses. Our future work will explore how to better use the CIDO ontology for more advanced applications.

## Supporting information

**S1 File. Detailed lists of natural and laboratory human coronavirus hosts and transgenic mouse models.** This is an Excel file with two spreadsheets, with Spreadsheet 1 providing detailed information on natural and laboratory human coronavirus hosts, and Spreadsheet 2 providing information on transgenic mouse models.
(XLSX)

**S2 File. SPARQL query of CIDO ontology for identifying specific SARS-CoV-2 hosts.**
(DOCX)

## Acknowledgments

We thank the anonymous reviewers for their valuable comments and suggestions on an earlier version of this paper. Noted that an early version of the paper was published as a preprint in *bioRxiv* with doi: https://doi.org/10.1101/2023.02.05.527173.

## Author Contributions

**Conceptualization:** Yang Wang, Yongqun He.

**Data curation:** Yang Wang, Muhui Ye, Fengwei Zhang.

**Formal analysis:** Yang Wang.

**Funding acquisition:** Xianwei Ye, Yongqun He.

**Investigation:** Yang Wang.

**Methodology:** Yang Wang, Hong Yu.

**Project administration:** Yongqun He.

**Resources:** Muhui Ye, Yongqun He.

**Software:** Yongqun He.

**Supervision:** Xianwei Ye, Yongqun He.

**Visualization:** Fengwei Zhang.

**Writing – original draft:** Yang Wang, Muhui Ye.

**Writing – review & editing:** Zachary Thomas Freeman, Hong Yu, Xianwei Ye, Yongqun He.

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
