## [Decision Letter · Decision Letter 0]

1 Jun 2023

PONE-D-23-12232Taxonomical and ontological analysis of verified natural and laboratory human coronavirus hostsPLOS ONE

Dear Dr. Wang,

Thank you for submitting your manuscript to PLOS ONE. After careful consideration, we feel that it has merit but does not fully meet PLOS ONE’s publication criteria as it currently stands. Therefore, we invite you to submit a revised version of the manuscript that addresses the points raised during the review process.

We look forward to receiving your revised manuscript.

Kind regards,

Sheikh Arslan Sehgal, PhD

Academic Editor

PLOS ONE

“Yes, this research was supported by Youth Found of Guizhou Provincial People's Hospital of China, GZSYQN[2019]09,”

“This research was supported by Youth Found of Guizhou Provincial People's Hospital of China, GZSYQN[2019]09, Guiyang Science and Technology Bureau Science and Technology Major Special Plan, Guizhou Provincial People's Hospital Public Health and Epidemic Prevention and Control Series Research Contract [2020] -4-1, and a bridge fund (to YH) at the Unit for Laboratory Animal Medicine in University of Michigan Medical School.”

“Yes, this research was supported by Youth Found of Guizhou Provincial People's Hospital of China, GZSYQN[2019]09,”

Reviewers' comments:

Reviewer's Responses to Questions

**Comments to the Author**

1. Is the manuscript technically sound, and do the data support the conclusions?

Reviewer #1: Partly

Reviewer #2: Partly

2. Has the statistical analysis been performed appropriately and rigorously? 

Reviewer #1: Yes

Reviewer #2: N/A

3. Have the authors made all data underlying the findings in their manuscript fully available?

Reviewer #1: Yes

Reviewer #2: Yes

4. Is the manuscript presented in an intelligible fashion and written in standard English?

Reviewer #1: Yes

Reviewer #2: No

5. Review Comments to the Author

Reviewer #1: 1.Line 29:

What is the relationship between host taxonomy and ontological analysis?

The significance of analyzing 37 different hosts is unclear. What is the implication of their belonging to therian mammals?

2.Line 37: Likely SARS-CoV-2 hosts ... ...

It is deemed unnecessary to explicitly state the handling method for them as it is standard protocol for data inclusion.

3.Line 38: mouse model with hACE2... ... Line 319: modified mice show significantly higher coronavirus infection rates ......

What is the significance of this conclusion? Is this solely due to the purpose of experimental design modeling? What new information is provided through the work presented in this article?

What is the main purpose of studying transgenic mice in this research? Is it possible that combining this type of information with knowledge of natural infections in a single ontology could lead to incorrect inferences?

How are transgenic mice represented in CIDO? Are they classified as a subclass of mice, or is a separate category established to represent these special organisms?

4.Line 42: What is the purpose of ultimately creating an ontology? Why is it important to analyze

host taxonomy first before constructing the ontology?

5.Line 59: It is critical to understand where coronaviruses originated and how they cause human outbreaks.

Has this question been addressed through the work presented in this article?

6.Line 67: the exact scope of the human coronavirus hosts and their transmissional relations still remain unclear.

Has the answer to this question become clear through the work presented in this article?

7.Line 85：suggesting that host susceptibility depends on many factors including genetic modification for coronavirus binding to host cells.

Since genetic modification is crucial, these data should be incorporated into the ontology.

8.Line 141: we extracted the hierarchical structure of various coronaviruses, and verified natural and laboratory animal hosts from the NCBI Taxonomy ontology ... ...

What’s relationship between the hierarchical structure of different types of viruses and the work “Taxonomical and ontological analysis of verified natural and laboratory human coronavirus hosts”? How the “verified”step processed?

9.Line 154: ACE2 Phylogenetic analysis is important,

What is the difference between analyzing only the ACE2 protein and analyzing the species phylogenetic tree? What additional information can be provided? Shouldn't a comprehensive analysis be conducted on all genes that may be associated with the viral infection process and host response?

How are the biological sequences for constructing the phylogenetic tree selected? In the Refseq database, there is a data entry for the ACE2 protein of a species, including the precursor, mature form, and isoforms. How did the authors consider these variations when constructing the phylogenetic tree? Why are there duplicate protein sequences in the tree shown in Figure 5?

10.Line 170: To demonstrate the usage of the CIDO representation ... ...

What is the contribution of this work to CIDO? Can CIDO incorporate more information? In "demonstrate the usage," the advantages of integration into CIDO should be showcased.

11.Line 192, 207:

Many of the findings in the paper are similar to those of other studies, which weakens the biomedical significance of these findings. The results and discussions in the paper are often mixed together and should be separated and clearly explained.

12. Line 237: this similarity is still insufficient to indicate that Malayan pangolins are intermediate hosts ... ...,

How can we determine if a species is an intermediate species by examining its taxonomical classification? What are the criteria for such determination?

13. Line 257：

Here, further analysis was conducted on verified laboratory animal hosts of humans. However, a clear logical line of reasoning was not established, and the purpose of this analysis was not clearly explained. What is the context of these analyses within the entire article?

14. Line 360: provides a demonstration of a SPARQL script used to find the number of verified organisms that are capable of being infected by SARS-CoV-2 virus through SPARQL query of knowledge stored in CIDO.

The demonstration provided is too simplistic. It is advisable to consider introducing one or more better demonstrations for advanced application by incorporating additional information from CIDO. The example is not sufficient to demonstrate the superiority of using ontology for knowledge representation.

15. It is recommended to use more standardized classifications for the Evidence column in Table 1 and Table 2, rather than extracting vague information from the original text.

Reviewer #2: This article, while interesting, is a literature review article, so does not describe novel research, but rather attempts to ascertain an exhaustive list of coronavirus hosts. While inclusion of a peer-reviewed article was dependent on those previous authors using experimental confirmation methods such as virus isolation, genomic sequencing, RT-PCR, and antibody neutralization assay, this article did not try to replicate any of these studies. Instead of saying this is a taxonomical ontology-based analysis, the authors must clarify whether this is secondary analysis of existing research or if novel analysis is being conducted, describe that more clearly in the Methods section.

Additionally, the article needs copy editing, as the English does not yet read as a publication ready article.

6. PLOS authors have the option to publish the peer review history of their article (what does this mean?). If published, this will include your full peer review and any attached files.

Reviewer #1: No

Reviewer #2: No

---

## [Author Response · Author response to Decision Letter 0]

6 Nov 2023

Responses to the Reviewers:

We appreciate the time and efforts of the reviewers who reviewed our manuscript. Their suggestions and comments are constructive and helpful. We have revised our manuscript by incorporating these comments accordingly as described below. A “track-change” supplemental file is also included to indicate all the changes in this revision compared to previous submission. It is noted that the italicized paragraphs below are the reviewer’s comments and our followed replies are in regular font style. 

Reviewer #1: 

1. Line 29: What is the relationship between host taxonomy and ontological analysis? 

The significance of analyzing 37 different hosts is unclear. What is the implication of their belonging to therian mammals? 

Reply: The host taxonomy analysis and ontology analysis are in a sense equivalent since here we primarily used NCBI Taxonomy Ontology (NCBITaxon). The NCBI taxonomy database provides the taxonomical structure of approximately 1 million taxonomical terms, which is difficult to extract a subset of animal species and form a hierarchical structure of these species. After we transfer the NCBI taxonomical structure to ontology (i.e., NCBITaxon), we can use tools such as Ontofox to easily generate a subset of the taxonomical structure that only includes the needed taxonomical terms. 

To avoid the confusion, we have changed the title from: “Taxonomical and ontological analysis of verified natural and laboratory human coronavirus hosts” 

To: “Ontology-based taxonomical analysis of experimentally verified natural and laboratory human coronavirus hosts and its implication for COVID-19 virus origination and transmission” 

 Furthermore, we have added the following text to the revised manuscript:

“In the informatics domain, ontology is a structured vocabulary that represents entities and relations among the entities in a specific domain using a human- and computer-interpretable format [13]. Many biological and biomedical ontologies have been developed and widely used. For example, the NCBITaxon ontology [14] is a taxonomy ontology developed based on the classification of various types of cellular organisms and noncellular self-replicating organic structures including viruses in the NCBI taxonomy database. The NCBI Taxonomy system provides a way to search the taxonomical information of specific animal types. However, it does not provide an automatic way to extract the hierarchical and integrative taxonomical information from a group of animal types. To address this issue, we could utilize the NCBITaxon ontology, which is derived from the NCBI Taxonomy system, and a specific tool (such as Ontofox [15] as used in this paper) to extract a small subset of the taxonomical hierarchy of related animals in an automatic and efficient way.” (Lines 74-85, Page 4). 

The significance of analyzing all the different hosts (including 37 natural hosts) is primarily to identify possible pattern(s) among these natural hosts and how the results could possibly infer the mechanisms of COVID-19 virus origination and transmission. Our study found that these natural hosts are all under the therian mammals, which is indeed a pattern that we would like to highlight. Based on our and other’s findings, we have also later proposed a MOVIE model in the Discussion section of the revised manuscript (see more later), which has been included in the revised Abstract and manuscript. 

2. Line 37: Likely SARS-CoV-2 hosts ... ... 

It is deemed unnecessary to explicitly state the handling method for them as it is standard protocol for data inclusion. 

Reply: This sentence has been deleted in the revised abstract as instructed. 

3. Line 38: mouse model with hACE2... ... Line 319（413）: modified mice show significantly higher coronavirus infection rates ...... What is the significance of this conclusion? Is this solely due to the purpose of experimental design modeling? What new information is provided through the work presented in this article? What is the main purpose of studying transgenic mice in this research? Is it possible that combining this type of information with knowledge of natural infections in a single ontology could lead to incorrect inferences? How are transgenic mice represented in CIDO? Are they classified as a subclass of mice, or is a separate category established to represent these special organisms? 

Reply: The purpose of adding the experimentally modified mice was to investigate the possible origination of COVID-19 virus. Our mouse model study has two findings: First, the mouse models with genetically modified human angiotensin-converting enzyme 2 (ACE2) or dipeptidyl peptidase-4 (DPP4) were more susceptible to virulent human coronaviruses with clear symptoms; Second, coronaviruses often became more virulent and adaptive in the mouse hosts after a series of viral passages in the mice. The first finding suggested that human ACE2 or DPP4 contributed to the human coronavirus infection and virulence. The second finding suggested that coronaviruses might have originated by increasing its virulence after a series of viral passages in an animal species. 

 Accordingly, we have modified the two sentences in the Abstract to the following:

“The mouse models with genetically modified human ACE2 or DPP4 were more susceptible to virulent human coronaviruses with clear symptoms, suggesting the critical role of ACE2 and DPP4 to coronavirus virulence. Coronaviruses became more virulent and adaptive in the mouse hosts after a series of viral passages in the mice, providing clue to the possible coronavirus origination.” (Lines 36-40, Page 2). 

Furthermore, these findings support our further proposal of a new MOVIE model (i.e., Multiple Organism Variations and Immune Evasion) to address how virus variations in animal hosts and the host immune evasion might have led to dynamic COVID-19 pandemic outcomes. The new MOVIE model has been added in the revised Discussion section. 

To represent transgenic mice in CIDO, we added a new class called ‘transgenic mouse’ under ‘mouse’, and all the specific transgenic mouse types are defined as descendent classes under the ‘transgenic mouse’. We have provided more details in the Results of the manuscript. 

4. Line 42: What is the purpose of ultimately creating an ontology? Why is it important to analyze host taxonomy first before constructing the ontology? 

Reply: Part of the explanation is described in our response to the reviewer’s first comment as seen above. In addition, we added the following text in the Discussion section:

 “Our ontological methods are two-fold. Firstly, our taxonomical analysis is an ontology based. Specifically, the whole taxonomical analysis performed in our manuscript was based on NCBI Taxonomy Ontology (i.e., NCBITaxon). After we mapped the animal types to specific ontological identifiers in NCBITaxon, we used the tool Ontofox to extract these animal names and the names of their specific ancestors, and then used the Protégé-OWL editor to build up the taxonomical trees as seen in the specific taxonomical figures. Secondly, we used the CIDO for modeling and representation of verified knowledge related to human coronavirus hosts. The extracted NCBITaxon ontological results were added to the CIDO ontology and become parts of the CIDO ontology. Furthermore, the CIDO ontology is extended to include other information such as transgenic mice and their transgenic modification and susceptibility to coronavirus infection. The incorporation of the known information in the CIDO make it feasible to develop advanced applications such as the SPARQL query, data integration and standardization, machine learning, natural language processing [18, 45, 86].” (Lines 551-563 in Page 30 in Discussion) 

5. Line 59: It is critical to understand where coronaviruses originated and how they cause human outbreaks. Has this question been addressed through the work presented in this article? 

Reply: We appreciate the reviewer’s comments and question. In the revision, we have proposed a new MOVIE model - “Multiple Organism Viral Variations and Immune Evasion”, which addresses where coronaviruses originated and how they cause human outbreaks. The model is developed based on the findings from our study and the knowledge learned from the other studies. We have provided the details in the Discussion section. This part is also copied below: 

“To further address the mechanism of the COVID-19 host origination and transmission, we propose a new MOVIE model - “Multiple-Organism viral Variations and Immune Evasion” based on the findings reported in this study and from the literature. The MOVIE model states that the complex dynamic interactions between COVID-19 viruses and hosts occurs in multiple host organisms in which the viruses undergo continuous genetic variations and immune evasion under various viral, host, and environmental conditions. The viral variation is the major mechanistic process, through which immune evasion occurs to achieve survival in the host. For the COVID-19 pandemic, our MOVIE model has the following tenets: 

(1)SARS-CoV-2 Origination: SARS-CoV-2 originated from continuous viral genetic mutations/variations in one or more host species that led to increasing viral virulence in the species. As shown in our annotated laboratory animal models (Table 3), the viral virulence turned to steadily increase in the same laboratory animal model in the early generations of viral passages in the host species. More evidence includes the early virulence increasing of SARS-CoV-2 variants including Alpha, Beta, and Delta strains. The sufficient genetic mutations in one host species might make the virus more susceptible to infect another species such as humans. We can further hypothesize that SARS-CoV-2 had gone a long time of hidden genetic variations without human notice, virulence increasing, and transmission in different animal species before the human outbreak observed by humans. 

(2)SAR-CoV-2 Viral Immune Evasion: SARS-CoV-2 viruses survive and evolve over time to achieve immune evasion in the hosts. Human coronaviruses conduct continuous genetic mutations and immune evasion under various host and environmental pressures. Numerous genetic variations in SARS-CoV-2 occurred during the coronavirus evolution from their continuous interactions with the hosts (84). Such variations likely make the virus achieve “immune escape”, i.e., escaping from the host immune system and then becoming more adaptive in the host (85, 86). 

(3)Multi-host COVID-19 Infections: Multiple COVID-19 hosts exist, and susceptible SARS-CoV-2 viral hosts are therian animals. The tenet is supported by our current findings that all experimentally verified natural, and laboratory human coronavirus hosts are under the Theria. Furthermore, the infection appears to be largely dependent on the SARS-CoV-2 S-protein:ACE2 complex binding capability, which can be estimated by the change in energy (ΔΔG) of the binding. Specific coronavirus genetic mutations might have occurred in the Spike gene to make the virus more capable of infecting the hosts by S-protein:ACE2 binding. 

(4)Spiral HCI Dynamics: The dynamic host-coronavirus interaction (HCI) is a complex system displayed as a spiral model in terms of its dynamic infection, transmission, and outcome manifestation. The “spiral” model of the complex HCI dynamics has at least two meanings: First, while the coronaviruses likely increase their virulence via genetic mutations to achieve immune evasion and host adaption at the early stage, the viral virulence likely decreases and the transmission likely increases over time due to intensive dynamic HCI processes at the population level. Correspondingly, the Delta variant is more pathogenic than the earlier variants, and the later Omicron variant is less pathogenic than the Delta variant. Second, while the first meaning provides a long-term effect, the short-time viral effects to the hosts are likely swing with unpredicted health outcomes. 

The Spiral HCI Dynamics tenet is supported by the mathematical Susceptible-Exposed-Infectious-Removed (SEIR) model for infectious disease dynamics [93, 94] and our proposed Spiral Symptom Occurrence Model Hypothesis [95]. Compared to the basic SIR model that has three groups (Susceptible, Infectious, and Recovered), the SEIR model adds the Exposed group for the latent period between being infected and becoming infectious. The SEIR model exhibits periodicity with open epidemic, and it identifies spiral waves over time that converge to the endemic equilibrium, and the model can also account for symptoms, transmission routes, and age [93]. The spiral symptom occurrence model hypothesis addresses the observed absence-presence-absence pattern of many special COVID-19 symptoms (e.g., loss of smell and taste) by spiral dynamics of host-coronavirus interactions through viral genetic variations, host responses, and viral immune evasion under various conditions [95]. The coronaviruses might have first randomly mutated into more virulent and transmissible variants leading to more death and special symptoms; however, given human responses due to natural infection or vaccination, those viral variants with increasing transmission rate survive better, and these fast-transmitting variants tend to show less virulent manifestation. One application of the spiral HCI model is to identify certain genetic mutations that are responsible for specific disease manifestations (e.g., death and loss of smell and taste), which may be present in some variants (e.g., Alpha and Delta variants) but not in the others (e.g., Omicron strains). 

Our proposed MOVIE model is aligned with and can be used to provide the mechanisms underlying many existing hypotheses. There have been three prevalent hypotheses regarding the evolutionary history of SARS-CoV-2 [96]. The first hypothesis states that COVID-19 viruses could have evolved in a “cryptically spread” way, which acts so that specific variant Omicron strains likely circulated in “covert transmission” among in a population(s) with insufficient viral surveillance and sequencing [97, 98]. The second hypothesis is that the Omicron variant may have evolved in a patient with long-term infection with COVID-19, who might have a compromised immune system [97, 98]. The single patient’s condition would allow the virus to adapt in a long term in the host. Many studies have reported that many viral mutations do exist in severely immune-compromised patients including those with AIDS and cancer [99-101]. A third hypothesis is that the Omicron variants might have accumulated mutations in their nonhuman hosts and then transmitted into humans through infection [99-101]. Several studies strongly argued that Omicron mutations were acquired from non-human hosts [97, 98]. Basically, the first “cryptically spread” hypothesis aligns with our SARS-CoV-2 Origination tenet that involves possible hidden genetic variations and transmission, which is accompanied with viral virulence increase at the early stage of outbreak. The second single-patient-incubation hypothesis can be explained by our viral immune evasion tenet that the single immunocompromised patient provides an ideal environment for the viral genetic variation and further immune evasion. The third nonhuman-to-human hypothesis can be explained by the Multi-host COVID-19 Infection tenet in our MOVIE model”

(Lines 600-679 in Pages 32-35 in Discussion) 

6. Line 67: the exact scope of the human coronavirus hosts and their transmissional relations still remain unclear. Has the answer to this question become clear through the work presented in this article? 

Reply: Our research focused on specifically surveying and analyzing the hosts of human coronaviruses, and our lab animal model study provided valuable suggestions to the transmissional mechanisms. Accordingly, we have proposed the MOVIE model (see above). With these, we believe the answer to the question raised by the reviewer has become clearer. 

7. Line 85：suggesting that host susceptibility depends on many factors including genetic modification for coronavirus binding to host cells. Since genetic modification is crucial, these data should be incorporated into the ontology. 

Reply: We appreciate the reviewer’s good suggestion. In the revision, we have added a new CIDO ontology representation figure (Figure 6) to further illustrate how CIDO represent the host susceptibility. The CIDO ontology has also included the description of genetic modification in each transgenic mouse model and many other places. For example, CIDO has represented different SARS-CoV-2 viral variants and the genetic variations and differences among different variants (Ref.: https://www.ncbi.nlm.nih.gov/pmc/articles/PMC9585694/). More genetic modification information will be added in the future. 

8. Line 141: we extracted the hierarchical structure of various coronaviruses, and verified natural and laboratory animal hosts from the NCBI Taxonomy ontology ... ... 

What’s relationship between the hierarchical structure of different types of viruses and the work “Taxonomical and ontological analysis of verified natural and laboratory human coronavirus hosts”? How the “verified” step processed? 

Reply: While the general objective of this study is to perform taxonomical and ontological analysis of verified natural and laboratory human coronavirus hosts, we first analyzed the hierarchical structure of various human coronaviruses in order to understand the relations among these human coronaviruses. The hierarchical analysis of these human coronaviruses allows us to know how these human coronaviruses are closely related, which provides a basis for our further analysis of the hosts of these human coronaviruses. 

 We have clarified the above meaning in the revised manuscript:

 “While this study aims to systematically analyze verified natural and laboratory human coronavirus hosts, the taxonomical analysis of these human coronaviruses allows us to know how these human coronaviruses are closely related, which provides a basis for our further analysis of the hosts of these human coronaviruses. Indeed, the taxonomical classification of the coronaviruses appears to be associated with the host species that these coronaviruses turn to infect. In general, Alpha- and Betacoronaviruses mainly infect mammalian species including humans, and Gamma- and Deltacoronaviruses primarily infect birds [6]. Note that although bats can fly, it is a mammalian species. Bat-borne betacoronaviruses are closely related and responsible for many human respiratory infections [44].” 

 (Lines 247-255 in Page 35; note the first sentence is newly added. The other sentences were in the original submission but had some minor updates.)

9. Line 154: ACE2 Phylogenetic analysis is important, What is the difference between analyzing only the ACE2 protein and analyzing the species phylogenetic tree? What additional information can be provided? Shouldn't a comprehensive analysis be conducted on all genes that may be associated with the viral infection process and host response? How are the biological sequences for constructing the phylogenetic tree selected? In the Refseq database, there is a data entry for the ACE2 protein of a species, including the precursor, mature form, and isoforms. How did the authors consider these variations when constructing the phylogenetic tree? Why are there duplicate protein sequences in the tree shown in Figure 5? 

Reply: The aim of the ACE2-based phylogenetic analysis is to support the identification of coronavirus binding capabilities in different species. The general species phylogenetic tree analysis would not be able to provide direct information on the coronavirus binding. The reason is that the SARS-CoV-2 virus infects animal species by binding to the ACE2 protein. The sequences of the ACE2 protein among different species vary, which may significantly affect the viral binding to host cells. 

 In the revision, we have added the following sentences:

 “Since ACE2 is critical to the coronaviral binding to host cells, an ACE2-based phylogenetic analysis has been used to compare coronavirus binding capabilities among different coronavirus hosts and predict possible coronavirus hosts [83].” (Lines 465-467, Page 26)

 Furthermore, as suggested by the reviewer, we have expanded our analysis to include 49 ACE2 sequences from animal species obtained from NCBI databases and from our natural and laboratory animal hosts resources (Fig. 5). These ACE2 sequences from different host species of human coronaviruses were discovered in the NCBI Protein Database, accessible at (https://www.ncbi.nlm.nih.gov). Our preference is to select the mature form from the corresponding species, followed by isoforms, with the precursor of the protein sequence being the last option. We have clarified these in our revision. 

10. Line 170: To demonstrate the usage of the CIDO representation ... ... What is the contribution of this work to CIDO? Can CIDO incorporate more information? In "demonstrate the usage," the advantages of integration into CIDO should be showcased. 

Reply: Coronavirus Infectious Disease Ontology (CIDO) is a database, so we need use to present the results in CIDO. CIDO can incorporate different types of information. More information is provided in supplementary file 2. 

11. Line 192, 207: Many of the findings in the paper are similar to those of other studies, which weakens the biomedical significance of these findings. The results and discussions in the paper are often mixed together and should be separated and clearly 

explained. 

Reply: Our paper started with the secondary collection and analysis of the results published in our studies. Therefore, it is true that many of the findings are similar to other studies. Meanwhile, we managed to make a comprehensive collection and annotation of all verified natural and laboratory animal hosts, which is a contribution by itself. Furthermore, we have conducted our own analyses and eventually raised the MOVIE model in the revision. We have made a significant effort to separate and clearly explain the results and discussions parts.

The first paragraph under the revised Discussion has summarized the contributions of this paper:

“The contributions of this article are multiple. Firstly, we systematically surveyed, identified, and collected 37 natural and 19 laboratory animal hosts of human coronaviruses (including SARS-CoV, SARS-CoV-2, MERS-CoV, and four less-virulent human coronaviruses) with experimental evidence. Secondly, our ontology-based taxonomical analysis found that all the verified natural and laboratory human coronavirus hosts are therian mammals, with the major category being Eutheria mammals (or placental mammals). The observation led us to hypothesize that human coronavirus hosts are therian mammals (see more discussion below). Thirdly, our mouse model meta-analysis identified 11 genetically engineered mouse models that were typically developed to express humanized ACE2 or DPP4 in order to become more susceptible to SARS-CoV, MERS-CoV, and SARS-CoV-2 infection. Furthermore, a series of viral passages in mice were also often implemented to increase the pathogenesis/virulence of the coronaviruses in the mouse models. Fourthly, we have modeled and represented these terms in the CIDO ontology and demonstrated their usage using SPARQL. Finally, based on the findings from this study, we propose a dynamic host-coronavirus interaction MOVIE model as described below.”

(First paragraph under Discussion). 

12. Line 237: this similarity is still insufficient to indicate that Malayan pangolins are intermediate hosts ... ..., How can we determine if a species is an intermediate species by examining its taxonomical classification? What are the criteria for such determination? 

Reply: The original Line 237 sentence is misleading; we have updated to Line 310:

“This similarity suggests that Malayan pangolin is likely an intermediate host directly involved in the current SARS-CoV-2 outbreak.” 

Also to note: the examination of taxonomical classification cannot determine if a species is an intermediate host. In this paper, we performed the taxonomical classification of all experimentally verified human coronavirus hosts, and our study showed that these verified human coronavirus hosts are all therian animals. Therefore, we hypothesize that all COVID-10 hosts are therian animals. We further performed an ACE2 phylogeny analysis to further investigate our hypothesis. 

13. Line 257：Here, further analysis was conducted on verified laboratory animal hosts of humans. However, a clear logical line of reasoning was not established, and the purpose of this analysis was not clearly explained. What is the context of these analyses within the entire article? 

Reply: We have significantly revised our paper by establishing a clear logical line of reasoning as the reviewer suggested. Our research focused on collection, annotation, and analysis of hosts of human coronaviruses (including its potential infected hosts) with our ultimate goal of deeply understanding the COVID-19 origination and transmission mechanisms among different host species. By summarizing our and others’ findings, we have proposed the MOVIE model in the Discussion section of the revised manuscript to address the question of COVID-19 origination and transmission mechanisms among different host species. We have also significantly revised the first paragraph under Discussion to summarize our contributions from this study. 

14. Line 360: provides a demonstration of a SPARQL script used to find the number of verified organisms that are capable of being infected by SARS-CoV-2 virus through SPARQL query of knowledge stored in CIDO. The demonstration provided is too simplistic. It is advisable to consider introducing one or more better demonstrations for advanced application by incorporating additional information from CIDO. The example is not sufficient to demonstrate the superiority of using ontology for knowledge representation. 

Reply: We appreciate the reviewer’s comment on the CIDO representation and usage. To better illustrate how CIDO represents the knowledge of SARS-CoV-2 hosts and host susceptibility, in the revision, we have added a new CIDO ontology representation figure (Figure 6). This new figure also shows the hierarchical structures from SARS-CoV-2 to the top-level term “Viruses” and from a representative host white-tailed deer up to therian mammals. Such ontological representation display would make readers have a better understanding of how CIDO represent COVID-19 host related knowledge. 

 Meanwhile, we have cited our previous CIDO papers, including the more recent comprehensive CIDO update paper: https://www.ncbi.nlm.nih.gov/pmc/articles/PMC9585694/, which introduces more demonstrations for advanced applications with the additional information incorporated in CIDO. We have added the following sentences in the revision to clarify:

 “By systematically incorporating COVID-19 knowledge in CIDO, we are able to develop more advanced applications, such as data standardization and integration, better mechanistic understanding of virulence and transmission, natural language processing (NLP) for clinical and basic mechanism research, and machine learning and drug cocktail design (16-18).” (Lines 87-91, Pages 3-4)

15. It is recommended to use more standardized classifications for the Evidence column in Table 1 and Table 2, rather than extracting vague information from the original text. 

Reply: Thank you for your kind suggestions. We have revised our Table 1 and Table 2. Using the same standardized classifications for the Evidence column, only left PCR, virus isolation, neutralizing antibody detection and so on.

Reviewer #2: 

This article, while interesting, is a literature review article, so does not describe novel research, but rather attempts to ascertain an exhaustive list of coronavirus hosts. While inclusion of a peer-reviewed article was dependent on those previous authors using experimental confirmation methods such as virus isolation, genomic sequencing, RT-PCR, and antibody neutralization assay, this article did not try to replicate any of these studies. Instead of saying this is a taxonomical ontology-based analysis, the authors must clarify whether this is secondary analysis of existing research or if novel analysis is being conducted, describe that more clearly in the Methods section. 

Reply: We agree with the reviewer that this paper did not have newly generated data; instead, this is secondary analysis of existing research. Specifically, this study started with the collection, annotation, and meta-analysis of experimentally verified human coronavirus host results with a special focus on COVID-19 hosts. We have clarified this by modifying the first sentence under Methods to:

“Instead of generating new data, this study began by identifying verified animal hosts of human coronaviruses from existing literature.” (Lines 113-114 in Page 6) 

Meanwhile, we did perform additional analysis of the collected data, esp. the ontology-based taxonomical classification of human coronaviruses and their hosts. 

We have also use the CIDO ontology as the platform for knowledge representation. Note that ontology is a major platform for knowledge representation, which is also an important branch of artificial intelligence (AI): https://en.wikipedia.org/wiki/Knowledge_representation_and_reasoning. 

Furthermore, we have added new features into the revision. We have systematically studied the Huanan Seafood Wholesale Market animals that were considered as the starting point of the COVID-19 outbreak in China. We also performed an advanced ACE2 phylogenetic analysis. Furthermore, based on our and other’s findings, we have proposed a MOVIE model in the Discussion section of the revised manuscript. 

Additionally, the article needs copy editing, as the English does not yet read as a publication ready article. 

Reply: We have made significant revisions to our manuscript and thoroughly edited the English copy to make it publication ready.

6. PLOS authors have the option to publish the peer review history of their article (what does this mean?). If published, this will include your full peer review and any attached files. If you choose “no”, your identity will remain anonymous, but your review may still be made public.

Reply: As authors of the paper, we are willing to publish the peer review history of their article.

---

## [Decision Letter · Decision Letter 1]

27 Nov 2023

Ontology-based taxonomical analysis of experimentally verified natural and laboratory human coronavirus hosts and its implication for COVID-19 virus origination and transmission

PONE-D-23-12232R1

Dear Dr. Wang,

We’re pleased to inform you that your manuscript has been judged scientifically suitable for publication and will be formally accepted for publication once it meets all outstanding technical requirements.

Kind regards,

Sheikh Arslan Sehgal, PhD

Academic Editor

PLOS ONE

Additional Editor Comments (optional):

Reviewers' comments:

Reviewer's Responses to Questions

**Comments to the Author**

1. If the authors have adequately addressed your comments raised in a previous round of review and you feel that this manuscript is now acceptable for publication, you may indicate that here to bypass the “Comments to the Author” section, enter your conflict of interest statement in the “Confidential to Editor” section, and submit your "Accept" recommendation.

Reviewer #1: All comments have been addressed

2. Is the manuscript technically sound, and do the data support the conclusions?

Reviewer #1: Yes

3. Has the statistical analysis been performed appropriately and rigorously? 

Reviewer #1: Yes

4. Have the authors made all data underlying the findings in their manuscript fully available?

Reviewer #1: Yes

5. Is the manuscript presented in an intelligible fashion and written in standard English?

Reviewer #1: Yes

6. Review Comments to the Author

Reviewer #1: The work is very comprehensive and systematic. I believe it is suitable for publication in the PLOS ONE journal.

7. PLOS authors have the option to publish the peer review history of their article (what does this mean?). If published, this will include your full peer review and any attached files.

Reviewer #1: No

---

## [Editor Report · Acceptance letter]

4 Dec 2023

PONE-D-23-12232R1 

Ontology-based taxonomical analysis of experimentally verified natural and laboratory human coronavirus hosts and its implication for COVID-19 virus origination and transmission 

Dear Dr. Wang:

I'm pleased to inform you that your manuscript has been deemed suitable for publication in PLOS ONE. Congratulations! Your manuscript is now with our production department. 

Kind regards, 

on behalf of

Dr Sheikh Arslan Sehgal 

Academic Editor

PLOS ONE